# Deep proteomic analysis of obstetric antiphospholipid syndrome by DIA-MS of extracellular vesicle enriched fractions

Wenmin Tian[1,5], Dongxue Shi[1,5], Yinmei Zhang[2,5], Hongli Wang [1], Haohao Tang[1], Zhongyu Han[2], Catherine C. L. Wong [3,4], Liyan Cui [2✉], Jiajia Zheng [2✉] & Yang Chen [1✉]

Proteins in the plasma/serum mirror an individual's physiology. Circulating extracellular vesicles (EVs) proteins constitute a large portion of the plasma/serum proteome. Thus, deep and unbiased proteomic analysis of circulating plasma/serum extracellular vesicles holds promise for discovering disease biomarkers as well as revealing disease mechanisms. We established a workflow for simple, deep, and reproducible proteome analysis of both serum large and small EVs enriched fractions by ultracentrifugation plus 4D-data-independent acquisition mass spectrometry (4D-DIA-MS). In our cohort study of obstetric antiphospholipid syndrome (OAPS), 4270 and 3328 proteins were identified from large and small EVs enriched fractions respectively. Both of them revealed known or new pathways related to OAPS. Increased levels of von Willebrand factor (VWF) and insulin receptor (INSR) were identified as candidate biomarkers, which shed light on hypercoagulability and abnormal insulin signaling in disease progression. Our workflow will significantly promote our understanding of plasma/serum-based disease mechanisms and generate new biomarkers.

[1] Department of Biochemistry and Biophysics, Center for Precision Medicine Multi-Omics Research, School of Basic Medical Sciences, Peking University Health Science Center, Beijing 100191, China. [2] Department of Laboratory Medicine, Peking University Third Hospital, Beijing, P R China. [3] Department of Medical Research Center, State Key Laboratory of Complex Severe and Rare Diseases, Peking Union Medical College Hospital, Chinese Academy of Medical Science & Peking Union Medical College, 100730 Beijing, China. [4] Tsinghua University-Peking University Joint Center for Life Sciences, Peking University, 100084 Beijing, China. [5] These authors contributed equally: Wenmin Tian, Dongxue Shi, Yinmei Zhang. ✉email: cliyan@163.com; gingkoblue@163.com; chenyang1816185048@bjmu.edu.cn

Proteins in the plasma/serum reflect the physiology of an individual. The size of the plasma/serum proteome is about 10,000 proteins, potentially comparable to cells and tissues[1,2]. Mounting evidence from proteomics or other disciplines suggests that the abundances of proteins present in plasma/serum change in ways that are indicative of many human diseases[3]. Thus, proteins in the plasma/serum hold the promise of a revolution in non-invasive diagnostic/prognostic tests and therapeutic monitoring. However, analyzing plasma/serum is challenging due to the complexity of the proteome and the wide range of protein abundance, up to 12 orders of magnitude.

Immuno-depletion of high-abundance plasma/serum proteins followed by MS-based analysis has successfully increased the depth of identification to a total of 1400 proteins, mainly consisting of secretory proteins[4]. Recent progress in analyzing plasma proteins enriched by the formation of a protein corona on the surface of nanoparticles improved the depth to about 2000 proteins[5]. An antibody-based approach with disease-specific bias identified up to 2923 proteins from plasma[6]. However, mining the proteome of such an important sample as plasma/serum has not been satisfactory. Unbiased biomarker discovery requires much greater depth in order to lead to novel disease pathways and potential biomarkers. Origin traceability and functional hints for disease-specific proteins identified from plasma/serum will further expand the role of the plasma/serum proteome in the exploration of disease mechanisms.

The proteome of extracellular vesicles (EVs) constitutes a large portion of the plasma/serum proteome, making it a promising candidate for liquid biopsies. EVs are cell-derived membrane-bound vesicles with heterogeneous contents, including genetic materials, proteins, lipids, and small metabolites. EVs facilitate tissue communications, especially long-term signal transduction, which is crucial in both physiological and pathological conditions[7]. Although released externally, EVs resemble their cell origin to some extent, a phenomenon called "tissue leakage". Based on size, EVs can be categorized into small EVs (SEVs; mostly exosomes) and large EVs (LEVs; microparticle)[7]. Exosomes are vesicles of 30–150 nm in diameter, originating from endocytic pathways[8]. Microparticles, which are 100 nm–1 μm in size, are derived from the plasma membrane and mainly contain cytoplasmic components[9].

EVs are important functional units in plasma/serum, and they contribute to a good portion of the plasma/serum proteome. Databases such as ExoCarta or Vesiclepedia present a very broad range of protein compositions of EVs, accumulated over the years.

Proteomic studies of circulating EVs under disease conditions were initiated in 2004[10]. EVs were studied using targeted protein-specific (e.g., via immunoassays) and untargeted LC–MS/MS proteomics strategies. Over the last two decades, a distinct pool of proteins was identified to be disease-specific, and this was successfully validated at the tissue level[11–16]. The study of circulating EVs with information about their origin and function has steered plasma/serum proteomics in a new direction. Depth of identification will be the rate-limiting factor for discovering biomarkers and exploring mechanisms. Extensive fractionation or extending the gradient is an option to improve the depth, but this can also magnify analysis time and limit throughput, and these drawbacks are not compatible with clinical cohort studies[11,17].

Here, we established a workflow for deep proteome profiling of LEVs and SEVs enriched fraction by UC plus 4D-data independent acquisition (DIA) mass spectrometry. This workflow significantly improved the depth of identification with high throughput, which is very important for biomarker discovery. EVs derived from serum are largely composed of cellular proteins and membrane proteins[11,18]. This will provide insights into disease pathology in clinical studies. We demonstrated the utilization of the workflow in a pilot study of OAPS, and we discovered that the proteomes of EV-enriched fractions yield novel biomarker candidates and information about specific disease pathways, including coagulation and insulin signaling.

## Results

**Deep coverage of serum proteomes through circulating EVs analysis.** We built up a procedure for in-depth characterization of the serum EV proteome. LEVs were isolated by $20,000 \times g$ centrifugation followed by 5–30% Iodixanol density-gradient centrifugation. SEVs were isolated by $100,000 \times g$ centrifugation followed by 0.25–2.5 M sucrose density-gradient centrifugation. All products have gone through extensive washing to yield high-purity LEVs (HPLEVs) or high-purity SEVs (HPSEVs). Secondly, the HPSEVs and HPLEVs were visualized by TEM and further characterized by immunoblotting for known EV biomarkers as well as EV-negative markers (Fig. 1b)[19]. The diameters, measured by NanoSight technology (NTA), were about 140 nm for HPSEVs and 250–500 nm for HPLEVs, in accordance with previous reports[20] (Fig. 1c). Thirdly, both HPLEVs and HPSEVs were lysed and digested in solution and subjected to 4D-DIA mass spectrometry. HPLEVs or HPSEVs were analyzed in a single run without any further protein or peptide fractionation, and 5636 and 4145 proteins were identified, respectively (Supplementary Data 1 and 2). This is comparable with the proteome size of EVs purified from conditioned medium[21]. The dynamic range of the HPLEV and HPSEV proteome showed enrichment of EV marker proteins as well as enrichment of low-abundance proteins (Fig. 1d). The HPSEV proteome contained 91 out of the top 100 exosome proteins from ExoCarta database. 3585 HPSEV proteins were in Vesiclepedia, and 560 were novel. Compared with HPSEVs or serum depleted of high-abundance proteins (immunodepleted serum), HPLEVs had an even broader protein composition: 4652 HPLEV proteins were in Vesiclepedia, and 1011 were novel (Fig. 1e). Most interestingly, subcellular localization analysis revealed that the identified proteins largely originated from cellular organelles and the plasma membrane and only 10% were secreted proteins (Fig. 1f), which is reminiscent of cell and tissue proteomes. Thus, serum EVs contain a large repertoire of proteins, potentially reflecting tissue conditions.

**Establishing a workflow for proteomics of serum SEV and LEV-enriched fractions.** Our data from HPLEVs and HPSEVs provide confidence that circulating EVs contain abundant proteins, among which are tissue leakage proteins. However, differential centrifugation followed by overnight density-gradient centrifugation limits throughput. To set up a workflow for cohort studies, we replaced the density-gradient centrifugation with two more washing steps to maximize the removal of contaminating proteins (Fig. 2a). We applied this workflow to serum samples from 6 healthy individuals (1 ml per sample), and we obtained washed crude LEVs (WCLEVs) and washed crude SEVs (WCSEVs). MS analysis of both WCLEVs and WCSEVs was achieved in a single run in 2 h. A total of 4000/1542 proteins were identified in WCLEVs/WCSEVs, with 2616/1070 proteins reproducibly identified in all 6 samples (Fig. 2b) (Supplementary Data 3 and 4). This compares favorably with 659 proteins reproducibly identified in all 6 immunodepleted serum samples[4] (Fig. 2b) (Supplementary Data 5).

Next, we compared data from the purification procedures in Fig. 1a and Fig. 2a. 3883 proteins were identified in both WCLEVs and HPLEVs from Fig. 1a, accounting for 69% of the HPLEV proteome and 97% of the WCLEV proteome (Fig. 2c). Similarly, 1454 proteins were identified in both WCSEVs and

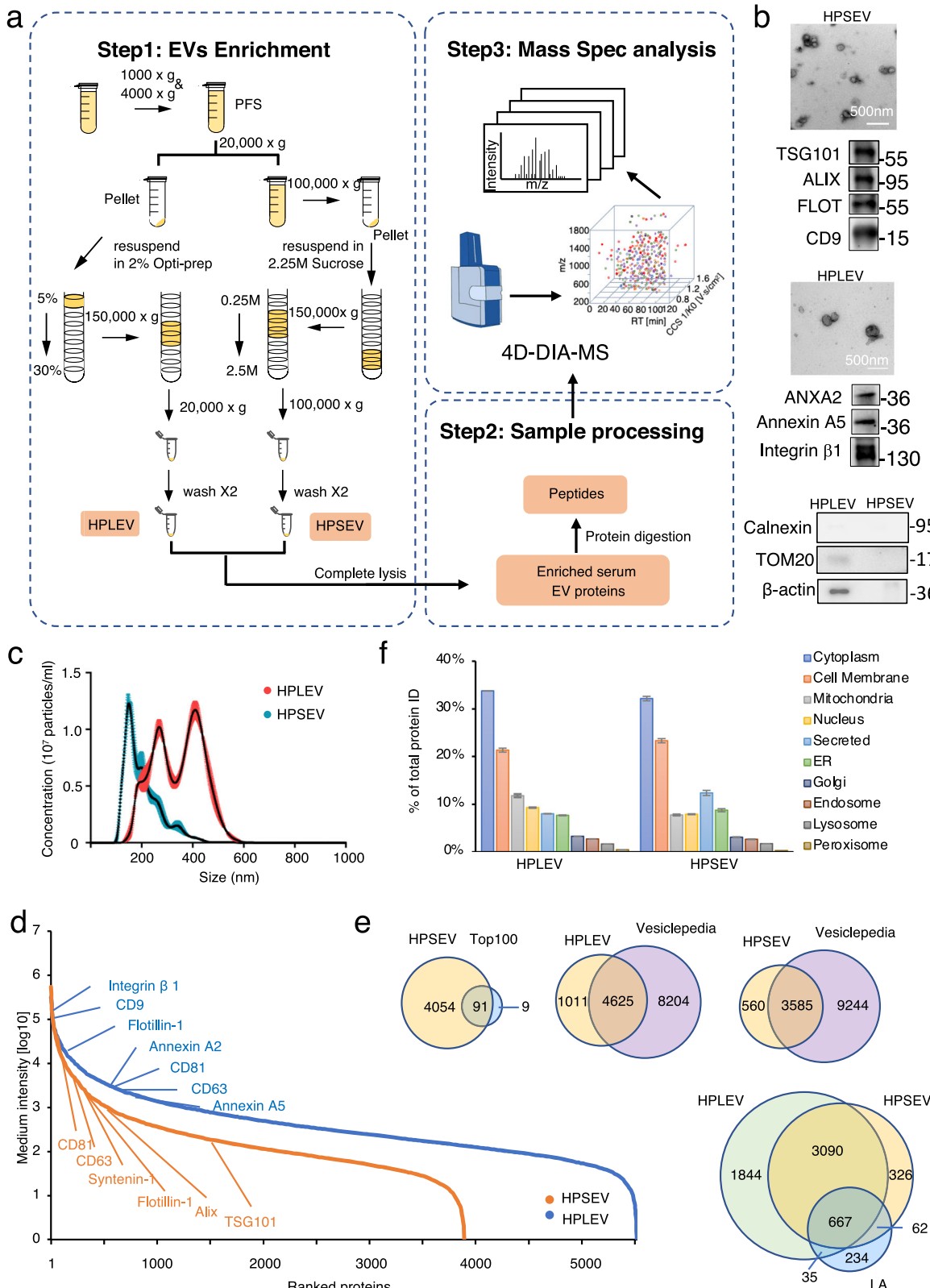

HPSEVs, accounting for 35% of the HPSEV proteome and 94% of the WCSEV proteome (Fig. 2c). For the procedure in Fig. 2a, both WCLEVs and WCSEVs yielded significantly more identified proteins than immunodepleted serum (Fig. 2c). The dynamic range for WCLEV and WCSEV (Fig. 2d) is similar to that for HPLEV and HPSEV (Fig. 1d). Protein localization analysis showed that a large proportion of both WCLEV and WCSEV proteins are cytoplasmic, membrane and organelle proteins (Fig. 2e). Thus, the WCLEVs/WCSEVs analyzed in our workflow contain mostly EV proteins, which are largely representative of the proteins found in HPLEVs and HPSEVs.

Comparing the proteomes of WCLEVs, WCSEVs, and immunodepleted serum, there were more drug targets, surface proteins, and transcription factors in WCLEVs and WCSEVs

**Fig. 1 Deep coverage of serum SEV and LEV proteomes. a** Illustration of the workflow for preparing high-purity LEVs/SEVs and performing mass spectrometry for proteomic analysis of serum EVs. The workflow includes three steps: (1) EVs enrichment, (2) Sample processing, and (3) Mass Spec analysis. HPLEVs high-purity LEVs, HPSEVs high-purity SEVs, PFS platelet free serum. **b** Characterization of HPLEVs and HPSEVs by TEM and immunoblotting for the indicated markers. **c** Representative NTA traces of HPLEVs and HPSEVs. The x-axis represents the diameters of the vesicles, and the y-axis represents the concentration ($10^7$ particles/ml) of the vesicles (n = 3 independent captures). **d** Relative abundance distribution of the proteins identified in serum HPLEVs and HPSEVs. Selected markers for LEVs (blue) and SEVs (orange) are marked on the curves. **e** Venn diagrams showing the overlapping proteins of the indicated samples or databases. Top 100 (ExoCarta), Vesiclepedia database, LA, immunodepleted serum. **f** Percentage of proteins from each subcellular localization identified in HPLEVs and HPSEVs.

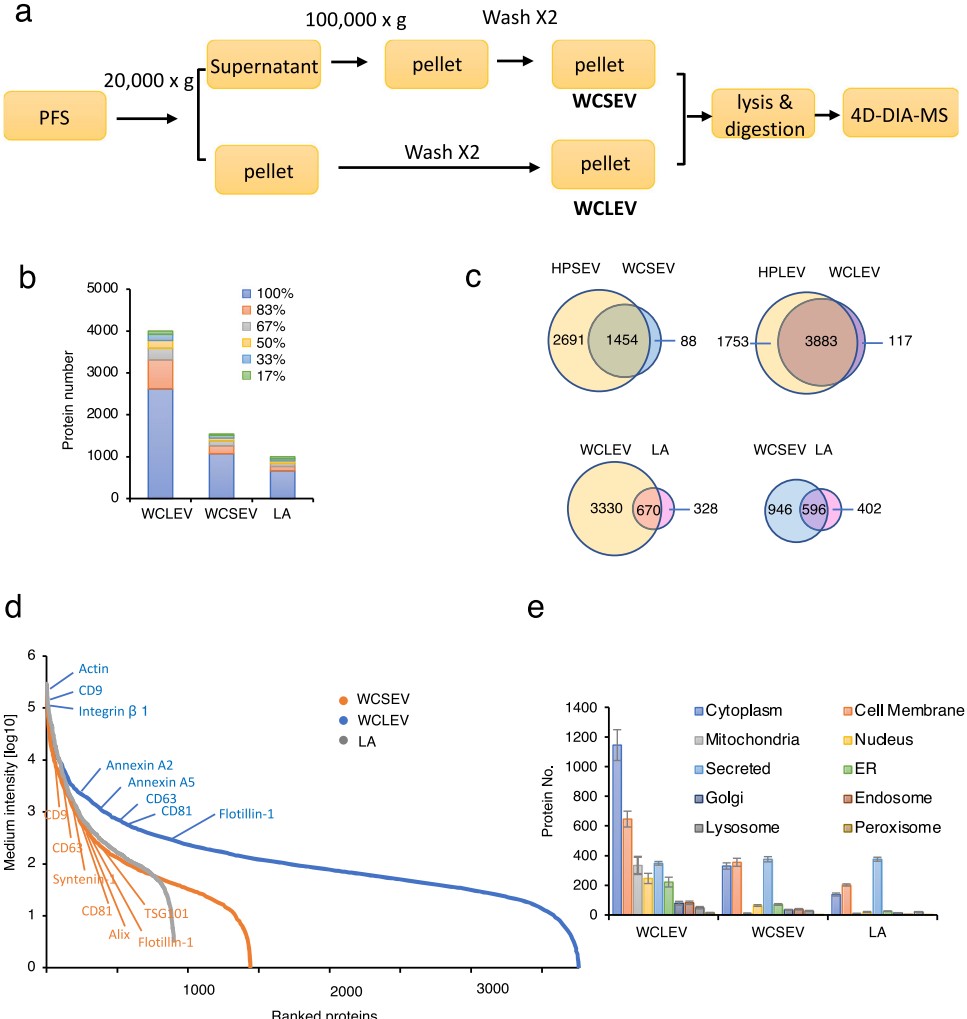

**Fig. 2 Establishing a workflow for proteomics of serum SEV and LEV enriched fractions. a** Illustration of the workflow for DIA-MS-based proteomics analysis of SEVs and LEVs from 1 ml serum of large clinical cohorts. WCLEVs washed crude LEVs; WCSEVs washed crude SEVs. **b** Number of proteins identified in WCLEVs, WCSEVs, and immunodepleted serum from 6 healthy individuals. Proteins identified in 100%, 83%, 67%, 50%, 33%, and 17% of the samples are labeled in the indicated colors. LA, immunodepleted serum. **c** Venn diagrams showing the overlapping proteins in the indicated samples. **d** Relative abundance distribution of the proteins in WCLEVs, WCSEVs, and LA. Selected markers for LEVs (blue) or SEVs (orange) are marked on the curves. LA: immunodepleted serum. **e** Proteins numbers from each subcellular localization identified in WCLEVs, WCSEVs and LA. LA, immunodepleted serum.

(Fig. 3a). It is worth noting that the WCLEV and WCSEV proteomes include almost the same number of FDA-approved biomarkers as immunodepleted serum (Fig. 3a), possibly because most of the existing FDA-approved biomarkers were discovered as canonically secreted proteins. For understanding disease mechanisms, secreted proteins are of interest, and EV-mediated proteins may also be highly relevant. We believe that in the near future, proteins identified from serum EVs will be an important source of biomarkers.

We also mapped functional annotations (GOBP, KEGG) and compared the enriched annotations in the proteomes of WCLEVs, WCSEVs and immunodepleted serum (Fig. 3b, c). WCLEVs and WCSEVs possessed significantly more functional annotations than immunodepleted serum. Moreover, WCLEVs covered more functional annotations than WCSEVs and immunodepleted serum (Fig. 3b). In addition, a large number of KEGG and GOBP pathways enriched in WCLEVs or WCSEVs were not significant or were even depleted in immunodepleted

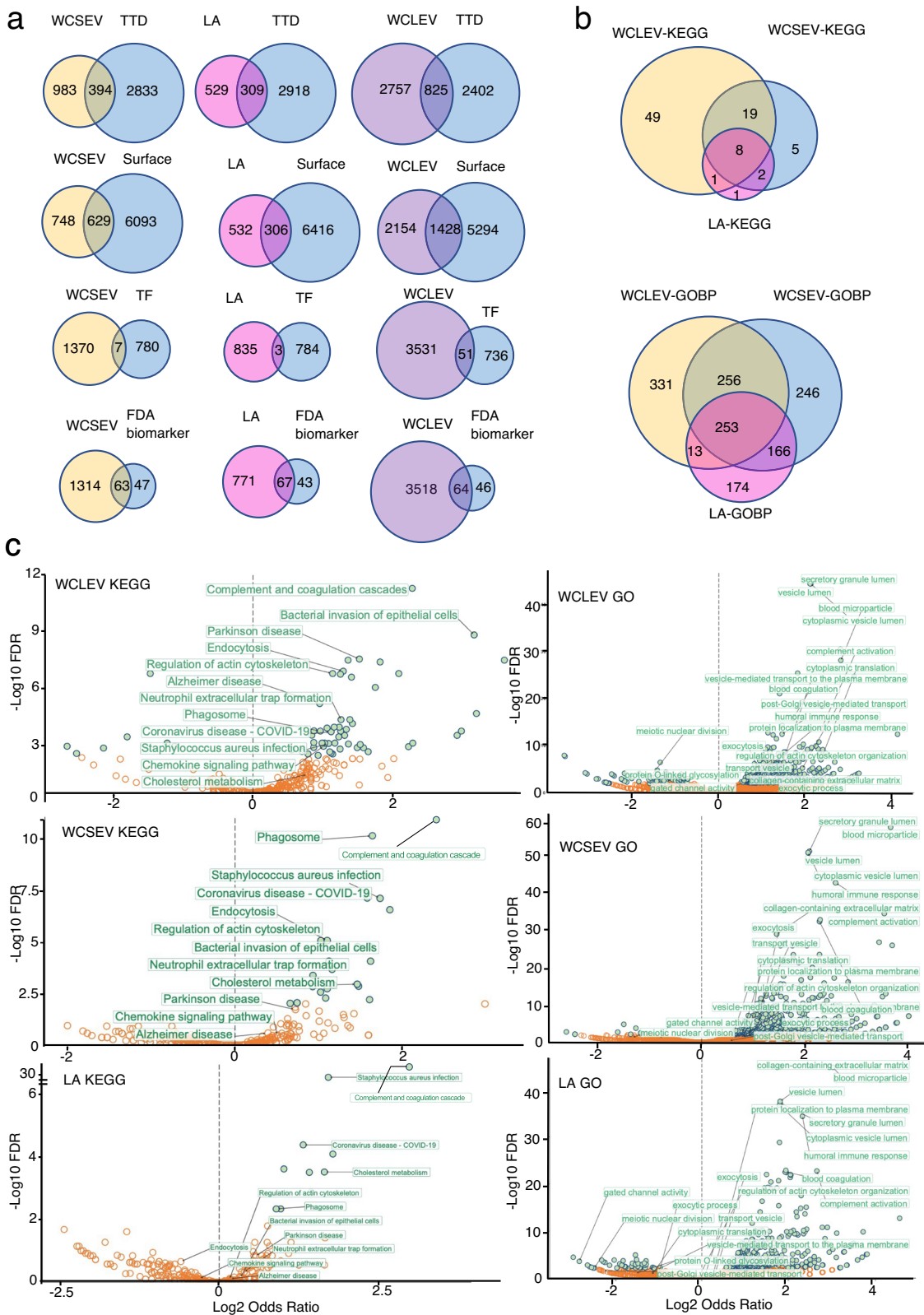

serum (Fig. 3c). This is consistent with the fact that the EV proteome represents a different pool of proteins compared to immunodepleted serum (Fig. 3c).

**Application of the workflow for proteomics of serum SEV and LEV enriched fraction to study OAPS.** To illustrate the

performance of the Fig. 2a workflow in a large human disease cohort, we performed deep proteome profiling of WCLEVs and WCSEVs from the serum of subjects with obstetric antiphospholipid syndrome (OAPS). Antiphospholipid syndrome (APS) is an autoantibody-induced thrombophilia whose hallmarks are recurrent thrombosis and pregnancy complications. So far, the pathogenesis lies mainly on endothelial cells, monocytes, platelets,

**Fig. 3 Property of proteomics of serum SEV and LEV enriched fractions from healthy individuals. a** Venn diagrams showing the overlaps between the proteomes of WCSEV/LA/WCLEV and proteins in the Therapeutic Target Database (TTD) (http://db.idrblab.net/ttd/) (top); human cell surface proteins (second row); proteins in the TRRUST transcription factor database (https://www.grnpedia.org/trrust/) (third row); and FDA-approved biomarkers (bottom row). The proteins analyzed are those identified in >67% of each sample. **b** Venn diagrams showing the overlapping enriched KEGG and GOBP pathways for the proteins identified in WCLEV, WCSEV, and LA (immunodepleted serum). **c** Volcano plots depicting annotation enrichment analysis (Fisher's exact test) for functional pathways (GOBP and KEGG) of proteins detected in each sample in comparison to the database. Enriched = log2 odds > 0; depleted = log2 odds < 0. Blue circles indicate pathways with a Benjamini–Hochberg (B.H.) false discovery rate (FDR) < 1%. Enriched annotations are labeled in green. Selected depleted annotations are depicted in orange. LA, immunodepleted serum.

and complement in the induction of thrombosis and fetal death in antiphospholipid syndrome[22]. OAPS subsets are featured by recurrent early miscarriages, fetal death at or beyond 10 weeks of gestation, and early delivery due to severe preeclampsia or placental insufficiency[23]. We enrolled OAPS patients and age/gender-matched healthy controls (WCLEV: Healthy controls, $n = 24$, age = 33.25 ± 4.93; OAPS, $n = 20$, age = 32.15 ± 3.79; WCSEV: Healthy controls, $n = 18$, age = 33.72 ± 4.86; OAPS, $n = 20$, age = 32.15 ± 3.79) (Supplementary Data 10). WCLEVs and WCSEVs were prepared from 1 ml serum obtained from 44 cases. The obtained fractions were lysed, processed, and analyzed by 4D-DIA mass spectrometry. The data showed clear stratification of the different patient/control groups according to principal component analysis (PCA) (Fig. 4a). We identified a total of 4270 proteins and an average of 3500 proteins in the 44 WCLEV samples, and a total of 3328 proteins and an average of 2500 proteins in the 38 WCSEV samples (Fig. 4b) (Supplementary Data 7, 8 and 11). The median coefficient of variation (CV) of QC samples (pool of all samples) was 10% and 25% for WCLEVs and WCSEVs, respectively (Fig. 4c) (Supplementary Data 12). 1029 and 332 differentially expressed proteins (DEPs) were identified by comparing the proteomes of WCLEVs and WCSEVs in OAPS to healthy control (Fig. 4d). It is worth noting that only 60 DEPs overlap between WCLEVs and WCSEVs, which suggests that WCLEVs and WCSEVs carry different disease-related information (Fig. 4e).

KEGG and GO analyses were performed for patient vs. control DEPs from WCLEVs and WCSEVs. For OAPS vs control WCLEV DEPs, the enriched pathways included Chagas disease, focal adhesion, and platelet activation. For OAPS vs. control WCSEV DEPs, the enriched pathways included platelet activation, regulation of actin cytoskeleton, focal adhesion, and platelet activation. The color-coded boxes in Fig. 4f correspond to the OAPS-associated KEGG and GO pathways identified in platelets, fibrin clots, plasma and serum, and monocyte samples, respectively, as documented in previous literature[24]. Thus, proteomics of EV-enriched fractions can offer a multi-encompassing perspective on disease pathogenicity, comparable to insights derived from multiple other sample types.

Apart from known pathways from reported proteomic studies, we also identified several new pathways and related proteins that provide novel insights into this autoimmune disease. The WCLEV proteome suggests that the OAPS patients might be abnormal in their response to viral or bacterial infection and in cholesterol metabolism. The WCSEV proteome suggests that the insulin signaling pathway may be abnormal in the disease group (Fig. 5a). The enriched pathways and the participating proteins, as well as the relationships between the pathways and proteins, are shown in Fig. 5a. It has been reported that bacterial and viral infections precede APS[25]. It is also possible that OAPS patients show susceptibility or resistance to infection. It has long been reported that lipid metabolism, including cholesterol metabolism, is strongly related to OAPS[26,27]. Interestingly, serum total cholesterol and low-density lipoprotein were significantly increased, while high-density lipoprotein was significantly

decreased in patients with OAPS (Fig. 5b) (Supplementary Data 6).

To verify the results of the 4D-DIA MS experiment, we performed immunoblotting in another independent cohort of OAPS. We found that VWF, identified from the WCLEV proteome, is up-regulated in patients (Fig. 5c, d) (Supplementary Data 9), consistent with the hypercoagulability in patients of OAPS. The insulin receptor (INSR) was increased from the WCSEV proteome, and this was similarly validated in the other cohort of OAPS (Fig. 5c, d) (Supplementary Data 9, 13). Furthermore, the receiver operating characteristic (ROC) curve showed that VWF and INSR are effective in discriminating between the OAPS patients and healthy controls with an area of curve AUC of 0.726 and 0.799, respectively, and 0.896 with both proteins (Fig. 5e) (Supplementary Data 13). The combined biomarker candidate panel of VWF from the WCLEV and INSR from the WCSEV could identify OAPS patients with higher specificity and sensitivity compared to individual and other reported biomarkers.

## Discussion

EVs are circulating functional units. EVs contain soluble and membrane protein cargoes from the parent cells and tissues, and therefore, they carry a tissue-specific signature[28,29]. It has been well documented that EVs contain disease-causing or disease-related proteins[8,28]. Thus, circulating EVs are reservoirs of disease biomarkers. Many studies have profiled EV proteomics from serum to dissect a variety of diseases, including cancer, infectious diseases, neurodegenerative diseases, etc., and the discoveries were validated at the tissue level[11–13,30].

The vast majority of studies have to focus on the several hundreds of proteins identified from circulating EVs due to technical limitations. This has resulted in increasingly more published papers reporting the same or similar protein biomarkers or similar pathways in multiple disease conditions, severely impacting sensitivity and disease-specific biomarker development. Using MS-based technology, we and others have proved that there are about 4000–5000 proteins in EVs, both from serum and generated in vitro by cells[11,17,21,31,32]. Thus, a broad range of information carried by EVs has remained largely unexplored. One reason for the large discrepancy between the expectations arising from plasma/serum proteomics and the realities of understanding disease mechanisms and promoting clinical diagnostics lies in the insufficient depth of identification.

We developed a workflow of 4D-DIA MS-based protein profiling of washed crude LEVs and SEVs (WCLEVs and WCSEVs) from 1 ml serum. We confirmed that the proteins identified largely represented those from serum LEVs and SEVs purified by density-gradient centrifugation (HPLEVs and HPSEVs, Fig. 1). In cohort studies, we identified up to 4270/3328 proteins for WCLEVs/WCSEVs. This depth of proteomic data could be a source of potential biomarkers and reliable candidates for understanding disease mechanisms. In addition, WCLEVs and WCSEVs reflect distinct features of OAPS with only 60 overlapping DEPs. Our data reflect the properties of the two distinct

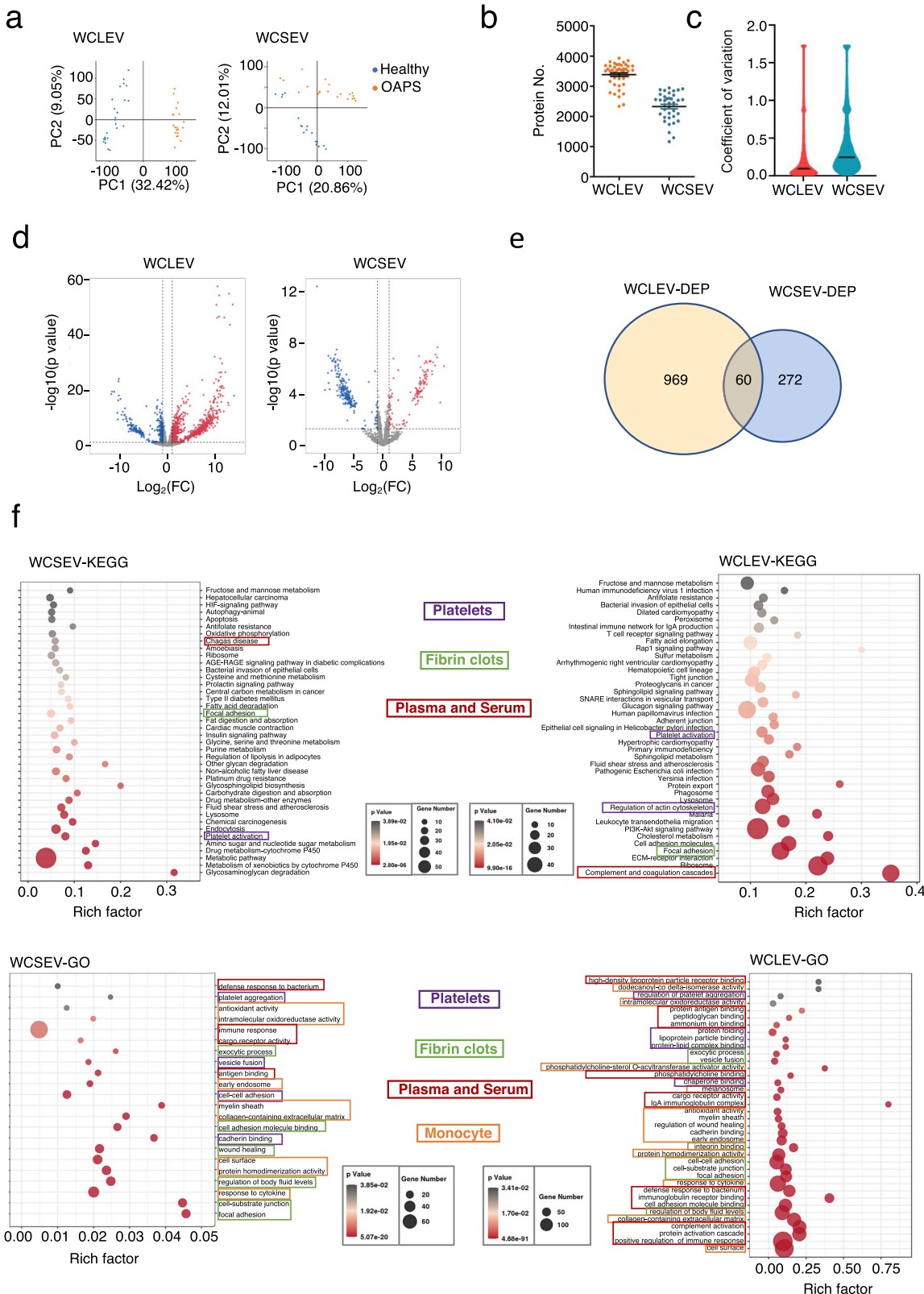

classes of EVs, which have distinct biogenesis mechanisms, compositions, and functions[33]. The depth of identification and the feasibility of our study will broaden the application of EV proteomics in biomarker discovery and disease mechanisms through samples from cohort studies.

So far, many plasma/serum EV proteomic studies have focused on SEVs, which are largely exosomes. Nowadays, more attention is being paid to LEVs, which are 100 nm–1 μm in size and contain a large variety of cytoplasmic components. For example, migrasomes are newly discovered organelles that detach from migrating cells to become EVs. Migrasomes are enriched in a combination of ligands, including chemokines, morphogens, and growth factors, and they function in organ morphogenesis, angiogenesis, and cellular homostasis[34–37]. Interestingly, mitochondria can also

**Fig. 4 Application of the workflow for proteomics of serum SEV and LEV enriched fraction in samples from OAPS disease cohort. a** PCA plot showing the sample clusters based on the proteomics data from WCLEVs (left) and WCSEVs (right). **b** Number of proteins identified in WCLEVs and WCSEVs. **c** Coefficient of variation (CV) of the proteomic data was calculated by the proteins quantified in three quality control (QC) samples. **d** Volcano plots showing the differentially expressed proteins (DEPs) in the comparison of WCLEVs (left) and WCSEVs (right) from OAPS patients and healthy controls. The red or blue colors show significantly increased or decreased proteins in the disease group with a cut-off value of fold-change > 2 or fold-change < 0.5; $p < 0.05$. **e** Venn diagrams showing the overlaps between differentially regulated proteins (DEPs) in the WCLEV and WCSEV samples from OAPS patients. **f** KEGG pathways and GO terms enriched in the differentially expressed proteins between OAPS patients and controls. Data from WCSEVs are shown on the left, and data from WCLEVs are shown on the right.

be cleared as large EVs through a secretory autophagy pathway[37]. Detached migrasomes have also been identified in serum, and their composition is distinct from exosomes[21]. It is also reported that large vesicles derived from cancer cells are detected in plasma/serum and function in the activation of pro-tumorigenic programs in target cells[38]. Thus, analyzing plasma/serum LEVs is as important as SEVs.

In this study, we noticed a larger discrepancy in protein identification between HPSEVs (4145) and WCSEVs (1542) than between HPLEVs (5636) and WCLEVs (4000). Our workflow to obtain WCLEV or WCSEV fractions from serum inevitably involves copurified proteins. It is highly possible that the $100,000 \times g$ centrifugation step in the preparation of WCSEV resulted in different highly abundant copurified proteins compared to the $20,000 \times g$ centrifugation step in the purification of WCLEV, which influenced their protein identification by mass spectrometry. For WCSEVs, the top 10 abundant proteins are mostly secreted proteins. They are immunoglobins, complement proteins, apolipoprotein, and albumin, and they account for more than 40% of the total protein amount. In contrast, the top 10 abundant proteins in WCLEVs are mostly intracellular proteins, such as actin, actin-regulated proteins, and integrins with relatively less albumin and immunoglobins. They account for about 30% of the total protein amount.

We demonstrated the feasibility of the workflow to differentiate between samples from OAPS patients and healthy controls. We identified known and new pathways and proteins as potential novel starting points for downstream studies[24]. Moreover, we used western blotting to validate the level of VWF and INSR in WCLEVs and WCSEVs from the serum of healthy individuals and patients in another cohort. VWF was identified both in the WCLEV and WCSEV proteomes as well as in immunodepleted serum, while its level was specifically increased in WCLEVs but not in the other two[24]. Hypercoagulation has been reported in OAPS patients[39]. Thus, VWF harbored in WCLEVs might play a significant role in this phenotype. According to the known mechanism of VWF biogenesis and secretion, we hypothesize that the source of VWF in WCLEVs may be ultra-large multimers VWF, surface-bound VWF on LEVs, or even LEV-encapsulated VWF that could indicate a new VWF secretion pathway[40]. Similarly, the increased level of INSR in WCSEVs was also validated. INSR is a cell membrane protein that is internalized intercellularly through binding to insulin. Canonically, INSR is trafficked to endosome–lysosome pathways or recycled back to the cell membrane. We found that INSR may also be harbored in WCSEVs. The increased level of INSR harbored in WCSEVs from OAPS serum may possibly hint at the abnormal INSR signaling. The relationship between OAPS and abnormal INSR signaling is rarely reported. However, mounting evidence indicates that increased insulin resistance is more prevalent in other autoimmune diseases compared to controls, for example, in systemic lupus erythematosus (SLE) and rheumatoid arthritis[41,42]. Our discoveries from analyzing the circulating EV proteome prompt us to rethink the function of circulating proteins in disease pathogenicity.

In addition, our workflow can be extended and tailored to other body fluids containing EVs (for example, plasma, urine, and cerebrospinal fluid) for deep, rapid, and accurate profiling of proteomes at low cost. We believe this workflow will boost the discovery of body fluid-based biomarkers and the study of disease mechanisms.

## Materials and methods

**Study population.** The participants were recruited from the Medical Examination Center of Peking University Third Hospital in 2022. All patients provided written informed consent. Female participants diagnosed with OAPS met the Sydney criteria as follows: (1) three or more consecutive spontaneous abortions prior to the 10th week of gestation; (2) one or more unexplained deaths of a morphologically normal fetus at or beyond the 10th week of gestation with normal fetal morphology documented by ultrasound or by direct examination of the fetus; (3) one or more premature births of a morphologically normal neonate before the 34th week of gestation because of eclampsia or severe preeclampsia or recognized features of placental insufficiency. Participants were excluded if there were maternal anatomical or hormonal abnormalities or if paternal and maternal chromosomal abnormalities caused abortion[23]. Meanwhile, healthy age-matched women were selected as controls. A total of 24 healthy individuals and 20 OAPS were enrolled (Supplementary Data 10) for mass spectrometry analysis. Demographics (age, gender, clinical characteristics) were obtained for all patients. In addition, 12 healthy individuals and 24 OAPS patients were enrolled for the validation study (Supplementary Data 9). The study was approved by the Ethics Committee of Peking University Third Hospital (ethical approval No. M2022836). All ethical regulations relevant to human research participants were followed.

**Serum sample preparation.** The participants were recruited from the Medical Examination Center of Peking University Third Hospital in 2022. In total, 3 ml blood samples of each participant for research purposes were collected in pro-coagulation vacuum tubes using standard venepuncture protocols. Serum was extracted by centrifugation for 10 min at $900 \times g$ and subsequently stored at $-80\,°C$ before use. All serum samples were prepared within 2 h after blood collection.

**Isolation of large and small extracellular vesicles (LEVs/SEVs) from serum**
*Isolation procedures for high-purity LEVs (HPLEVs).* A total of 20 ml cell-free serum was required for purification of one batch of HPLEVs (Fig. 1a). Large debris was removed by centrifugation at $1000 \times g$ for 10 min followed by $4000 \times g$ for 20 min to obtain platelet-free serum. The sample was diluted 1:6 in PBS to 120 ml for further centrifugation. The pellet was collected by centrifugation at $20,000 \times g$ for 45 min, then washed with PBS. Fractionation of the resuspended pellet was performed by Iodixanol-sucrose density gradient centrifugation, using Optiprep (Sigma-Aldrich, D1556) as the density medium. First, a step

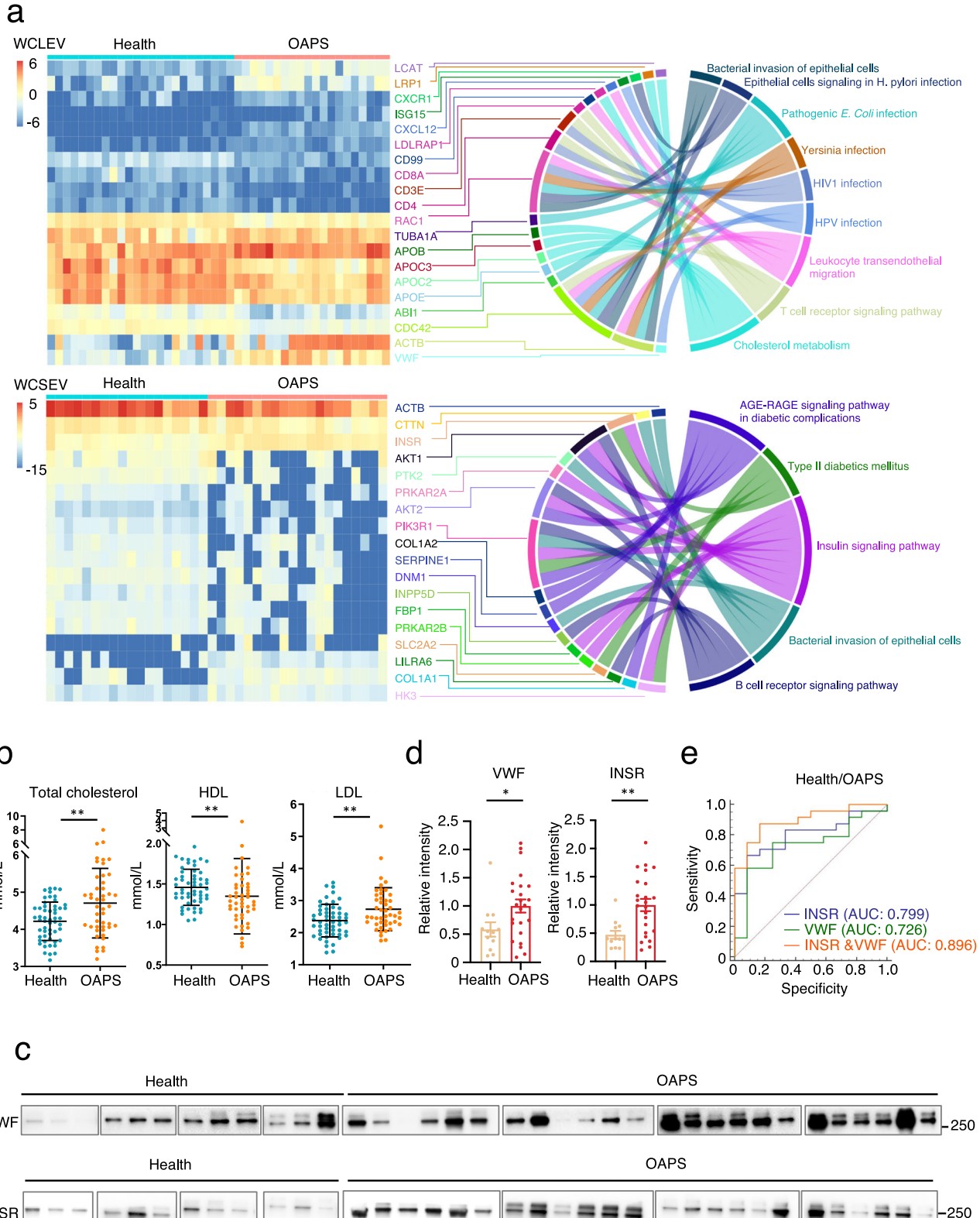

**Fig. 5 Verification of candidate biomarkers in EV enriched fraction by targeted assay. a** Sankey plots exhibiting the correlation between the enriched pathways and proteins from WCLEVs (upper panel) and WCSEVs (lower panel). **b** Plots showing significant differences in lipid levels between OAPS patients and controls. Asterisks indicate statistical significance based on unpaired two-sided Student's *t* test. **p < 0.01; **c** Immunoblots showing the levels of VWF in WCLEVs and INSR in WCSEVs from the serum of healthy controls and OAPS patients. EV samples with equal protein amounts were analyzed with indicated antibodies. **d** Quantitative analysis of the data in (**c**). The gray scale of each band representing the specific protein was extracted from the immunoblot images. Asterisks indicate statistical significance based on unpaired two-sided Student's *t* test. *p < 0.05; **p < 0.01. **e** ROC curves comparing the power of WCLEV VWF (solid green line), WCSEV INSR (solid blue line), and both of them (solid orange line) to differentiate between the healthy control and OAPS groups. The dotted red line is the reference line.

gradient was built starting from 30% (500 μl), followed by 25% (500 μl), 19% (500 μl), 15% (500 μl), 12% (500 μl), 10% (500 μl), 8% (500 μl), and 5% (500 μl). The sample (500 μl) was prepared in 2% Optiprep and layered on top. The gradient was centrifuged at 150,000 × g for 4 hr at 4 °C in an MLS-50 rotor (Beckman). Samples were collected from top to bottom (480 μl per fraction). Each fraction was mixed with the same volume of PBS (480 μl) and then centrifuged at 20,000 × g for 45 min. The pellets, containing high-purity large EVs (HPLEVs) at fractions 4 and 5, were washed with PBS and centrifuged again at 20,000 × g for 45 min. The HPLEV samples were compatible with western blot analysis, TEM, cryo-EM and mass spectrometry.

**Isolation procedures for high-purity SEVs (HPSEVs).** A total of 20 ml cell-free serum was required for purification of one batch of HPSEVs (Fig. 1a). Large debris and platelets were removed by centrifugation at 1000 × g for 10 min followed by 4000 × g for 20 min. Then, the sample was diluted 1:6 in PBS to 120 ml for further centrifugation. The supernatant was further centrifuged at 20,000 × g for 45 min to collect large EVs, as mentioned above. The serum supernatant after 20,000 × g centrifugation was further centrifuged at 100,000 × g for 70 min at 4 °C. The pellet was washed with 60 ml PBS, followed by a second step of ultracentrifugation at 100,000 × g for 70 min at 4 °C to collect the pellet. Sucrose density-gradient centrifugation was performed to further purify SEVs. Briefly, crude SEVs were resuspended in 500 μl of HEPES/sucrose stock solution (2.25 M sucrose, 20 mM HEPES/NaOH solution, pH 7.4). The suspension was overlaid with a step sucrose gradient (2.5, 2.0, 1.75, 1.5, 1.25, 1.0, 0.75, 0.5, 0.25 M sucrose, 20 mM HEPES/NaOH, pH 7.4, 500 μl). Crude SEVs (2.25 M sucrose, 20 mM HEPES/NaOH) were loaded in between the 2 M and 2.5 M sucrose steps. The gradient was spun at 150,000 × g at 4 °C for 4 h. Gradient fractions of 480 μl were collected from top to bottom, and the HPSEV pellets at fractions 8 and 9 were washed in PBS then subjected to a final ultracentrifugation step at 100,000 × g at 4 °C for 70 min. The high-purity SEV (HPSEV) samples were compatible with western blot analysis, TEM, cryo-EM, and mass spectrometry.

**Isolation procedures for washed crude LEVs/SEVs (WCLEVs/WCSEVs).** A total of 1 ml cell-free serum was used. Large debris and platelets were removed by centrifugation at 1000 × g for 10 min, followed by 4000 × g for 20 min. The supernatant was further centrifuged at 20,000 × g for 45 min to collect crude large EVs as the pellet. The large EVs were washed with PBS twice by centrifugation at 20,000 × g for 45 min. The previous supernatant was further centrifuged at 100,000 × g for 70 min to collect crude small EVs as the pellet. The small EVs were washed with PBS twice by centrifugation at 100,000 × g for 70 min. The resulting preparations were WCLEVs and WCSEVs.

**Nanoparticle tracking analysis.** A NanoSight NS 300 system (NanoSight Technology, Malvern, UK) was used to analyze the size distribution of EVs. The samples were diluted 150–3000 times with PBS without any nanoparticles to attain a concentration of $1–20 × 10^8$ particles per milliliter for analysis. Each sample was measured in triplicate, and each visible particle was recorded and tracked. EV numbers and size distribution were explored using the Stokes-Einstein equation. NanoSight NTA3.2 software was used for data analysis.

**Negative staining and transmission electron microscopy.** Purified HPLEV or HPSEV pellets were resuspended in 50–100 μl PBS, and then a 5 μl sample of each was mixed with the same volume of 2.5% glutaraldehyde (PB, pH 7.4) and fixed for 30 min

at room temperature. The sample was spread onto glow-discharged Formvar-coated copper mesh grids (Electron Microscopy Sciences, Hatfield) for about 5 min, then washed with water. The sample was then stained with uranyl acetate for 2 min. Excess staining solution was blotted off with filter paper, and copper mesh grids were washed with water. Post-drying, grids were imaged at 10–100 kV using a transmission electron microscope (H-7650).

**Western blot.** Purified EVs were lysed by RIPA (Thermo, Cat. No. 89901) buffer for 2 h, then centrifuged at 14,000 × g for 20 min. Supernatants with the same amount of protein were subjected to an SDS loading buffer and boiled for 10 min at 95 °C. Samples were separated on SDS-PAGE gels of an appropriate percentage according to the MW of target proteins, followed by electrophoretic transfer onto Nitrocellulose membrane. Membranes were blocked with 5% non-fat milk in TBST buffer and incubated with primary antibody overnight at 4 °C. Membranes were incubated with HRP-conjugated secondary antibody for 1 h at room temperature, and signals were detected with ECL Reagent (LABLEAD, E1050).

**Sample preparation for mass spectrometry.** EVs (5–20 μg) were lysed with 20 μl RIPA buffer (Thermo, Cat. No. 89901) on ice for 30 min and then centrifuged at 16,000 × g for 30 min. The supernatant was precipitated with 120 μl acetone overnight at −20 °C. The precipitate was resuspended in 20 μl of 8 M urea (500 mM Tris-HCl, pH 8.5) and sequentially treated with Tris (2-carboxyethyl) phosphine hydrochloride (10 mM) and iodoacetamide (25 mM) (Sigma) at room temperature. After digestion by trypsin (1 μg) overnight at 37 °C, the resulting peptides were desalted using a Monospin C18 column (GL Sciences) and redissolved in 0.1% formic acid.

**Peptide fractionation.** For spectral library construction, approximately 70 μg of mixed peptide was fractionated using a high-pH reversed-phase HPLC system (1290 Infinity, Agilent). Briefly, the mixed peptides were dissolved in Buffer A (20 mM ammonia, 2% ACN, pH 10.5) and loaded onto an Accucore™ C18 column (2.1 mm × 150 mm, 2.6 μm particles; ThermoFisher Scientific) using 56 min linear gradient of 5–80% of Buffer B (20 mM ammonia, 98% ACN, pH 10.5) at a flow rate of 0.4 mL/min. Then, the pooled fractions were combined into 12 fractions and dried completely by vacuum centrifugation. Dried peptides were dissolved in 10 μL Milli-Q water with 0.1% formic acid (FA) and then spiked with iRT peptides for chromatographic correction.

**Mass spectrometry.** All samples were separated using a nanoElute UHPLC spectrometer within 120 min on a reversed-phase C18 column (25 cm × 75 μm, 1.6 μm, IonOpticks) with an integrated Captive Spray Source. The elution gradient of B was linearly increased from 2% to 22% within 90 min, followed by an increase to 37% within 10 min at a flow rate of 300 nL/min at 50 °C (mobile phase A: 0.1% formic acid in water; mobile phase B: 0.1% formic acid in acetonitrile).

Spectral library generation was performed in data-dependent acquisition (DDA) mode. 12 eluted fractions were acquired on the Bruker timsTOF Pro mass spectrometer, which was operated in PASEF (Parallel Accumulation Serial Fragmentation) mode [1,2] with the following settings: mass range 100–1700 m/z, 1/K0 Start 0.6 Vs/cm², End 1.6 Vs/cm², Ramp time 100 ms, Capillary Voltage 1500 V, Dry Gas 3 L/min, Dry Temp 180 °C, PASEF settings: 10 MS/MS Frames (1.16 s duty cycle), charge range 0–5, Target intensity 20,000, Intensity threshold 2500, HCD collision energy 10 eV.

For sample analysis, diaPASEF (data-independent acquisition-Parallel Accumulation Serial Fragmentation) mode was used.

diaPASEF data were acquired at defined $50 \times 25$ Th isolation windows from m/z 400 to 1200. The collision energy was ramped linearly as a function of the mobility from 59 eV at 1/K0 1.6 Vs cm$^2$ to 20 eV at 1/K0 0.6 Vs cm$^2$.

**Generation of spectral libraries and DIA data analysis.** Spectral libraries were generated by Spectronaut version 16.0 with a UNI-PROT human database (only reviewed entries). All the parameters were default. DIA files were processed in default mode.

For the DIA-proteomic data presented in Fig. 1, a spectral library from the analyzed samples was used for DIA data identification. Meanwhile, for the proteomic data in Figs. 2–5, a hybrid of sample-specific spectral library and direct search-based library were employed for DIA data identification. For the spectral library generation, a 1% FDR was maintained at the PSM, peptide, and protein levels.

**Annotation-diversity analysis.** To determine which annotations are predominantly enriched in the EVs, we performed an annotation enrichment analysis using Fisher's exact test to compare proteins identified in the EVs. UniProt IDs (Majority protein IDs) were matched to a list of 5304 published plasma/serum proteins. Next, annotations from two databases, Gene Ontology (GO) and Kyoto Encyclopedia of Genes and Genomes (KEGG), were respectively matched to the protein groups based on Uniprot identifiers. Using Fisher's exact test, we determined enriched annotations comparing the population of proteins identified by the EVs within the reference database against the proteins that did not map into the EVs. Enrichment scores (log2 odds ratios) were calculated and plotted against the p-values (Fig. 4d). Annotations significantly enriched with a Benjamini–Hochberg FDR < 1% are indicated in blue. If log2 odds were infinite, the maximum/minimum log2 odds were used for drawing.

**Antibodies.** Primary antibodies were: anti-ALIX(1:1000), ab186429 (Abcam); anti-TSG101(1:1000), GTX70255 (GeneTex); anti-flotillin1(FLOT)(1:1000), 610820 (BD); anti-integrin β 1(1:1000), ab183666 (Abcam); ant-CD9(1:1000), ab223052 (Abcam); anti-Annexin A5(1:1000), #8555 S (CST); anti-Annexin A2(1:7000) (ANXA2), ab178677 (Abcam); anti-VWF(1:1000), 65707 (CST); anti-INSR(1:200), sc57342 (Santa Cruz Biotechnology); anti-Calnexin(1:1000), ab22595 (Abcam); anti-TOM20(1:1000), 11802-1-AP (Proteintech); anti-β-actin(1:1000), 8457 S (CST).

**Statistics and reproducibility.** Omicsbean software was used for data analysis, including data imputation, normalization, and principal component analysis (PCA). Fold-change of 2 or fold-change < 0.5 and p-value of 0.05 were used to filter differentially expressed proteins. Network visualization was performed using the ggplot228 packages and Cytoscape v.3.5.129 implemented in the Omicsbean workbench. $n = 5$ independent HPLEVs or HPSEVs were analyzed in Fig. 1. $n = 6$ independent WCLEVs or WCSEVs were analyzed in Fig. 2. $n = 44$ independent WCLEVs and $n = 38$ WCSEVs were analyzed in Fig. 4. $n = 36$ independent WCLEVs or WCSEVs were analyzed in Fig. 5.

**Reporting summary.** Further information on research design is available in the Nature Portfolio Reporting Summary linked to this article.

## Data availability

The experimental data that support the findings of this study have been deposited in iProX (integrated proteome resources) of ProteomeXchange with the accession code

PXD043290. The uncropped and unedited blot images were included in the Supplementary Figure.

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

## Acknowledgements

We are grateful to the Center for Precision Medicine Muti-omics Research of Peking University Health Science Center for the quantitative mass spectrometry analysis. We thank the Cell Facility in the Center of Biomedical Analysis of Tsinghua University for help with electron microscopy. We thank the Shanghai Omicsolution Co., Ltd. for their help in data analysis. This work is supported by the Ministry of Science and Technology of the People's Republic of China 2022ZD0212900, National Key Research and Development Program of China 2023YFF0613402, Research Funds from the Health@InnoHK Program launched by the Innovation Technology Commission of the Hong Kong Special Administrative Region, Key Clinical Projects of Peking University Third Hospital BYSYZD2021041 and Fundamental Research Funds for the Central Universities BMU2017YJ003.

## Author contributions

W.M.T. and Y.C. initiated the project. W.M.T., D.X.S., and H.H.T. conducted the proteomic experiment. Y.M.Z., Z.Y.H., L.Y.C., and J.J.Z. collected clinical samples and information. W.M.T., D.X.S., H.L.W., Y.C., C.C.W., Y.M.Z., Z.Y.H., L.Y.C., and J.J.Z. participated in data analysis. W.M.T. and Y.C. wrote the paper.

## Competing interests

The authors declare no competing interests.
