## [Peer Review File · Communications Biology]

Reviewers' comments:

Reviewer #1 (Remarks to the Author):

Dear Tian and co-authors,

In this study, you have performed a Deep proteomic analysis of circulating large/small extracellular vesicles in patients with obstetric antiphospholipid syndrome. You performed a fascinating and comprehensive study, and the article contains new information. However, in some parts, it could be written a little more carefully. Some information is missing, some numbers do not agree... It is often not clear whether the authors are comparing EVs isolated by different methods (WCLEV vs. WCSEV, HPSEV vs. WCSEV, HPSEV vs. HPSEV), all isolated from healthy individuals, or whether they are comparing healthy EVs with OAPS EVs... I suggest that the authors clearly state throughout the paper what they are comparing and which cohorts were used for which analysis.

Main comments...:

1. Title. There is no information about the study population; this may broaden the scientific audience that will read the paper.

2. Introduction. Add information about the size of the EVs at the end of the third paragraph (lines 83-84). I suggest that the first part of the results (lines 250-256) be moved to the introduction. This information should be available to readers at the very beginning.

3. Methods.

a. The order of the subtitles is illogical... it starts with antibody? Maybe: 1. study population, 2. sample preparation, 3. methods..., antibodies, 4. statistics...

b. The description of the western blot analysis is missing

c. Serum sample preparation. Serum samples should be prepared promptly after blood collection to minimize the degradation of EVs and possible alterations. The authors did not specify how long they stored the serum at -80°C before storage

d. Line 117: Serum was centrifuged at 3000 rpm... RPM units should be replaced by centrifugation force ($\times g$). As is stated in the rest of the manuscript.

e. I recommend using ($\times g$) units instead of just (g). For example, 1000 $\times g$, instead of 1000g

f. Please include '(Figure 1a)' in the paragraphs isolation procedures HPLEVs and HPSEVs. I suggest at the end of the first or second sentence, i.e.: "A total of 20 ml of cell-free serum was needed to purify a batch of HPLEVs (Figure 1a).

g. There is a missing paragraph about the participants included in the study, e.g.: 'Participants and clinical characteristics'. Please indicate that (or if) the workflow was first established using serum from healthy participants....(n=) and then add patient information. Please include information on inclusion criteria, how many patients were enrolled in the study, and which cohort was used for further analysis.

4. Results:

a. The first part of the results (lines 250-256) are actually not the results, as suggested this part should be in the introduction.

b. Lines 258-267 repeat the methods. This part should be much shorter. It lacks details about the participants... Therefore, later in line 273 should be: "...and 5636 and 4145 proteins were identified...." It is not clear if this was in patient sera or healthy sera.

c. Line 396. Add specific information, how many, what age...? (HBD, n=57, mean age 31.2 years) (OAPS, n=46, mean age 31.3 years)

d. Line 397: "44 cases" Which cases? In Table 6, there are 57 healthy individuals and 45 OAPS...?

5. Discussion: line 567. did the authors investigate if there are some OAPS patients with a history of thrombosis in their study? Maybe they could check if these patients have higher VWF values in their EVs.

Reviewer #2 (Remarks to the Author):

In this manuscript the authors describe a novel approach to establishing deep proteomic profiling of serum EVs via 4D-data independent acquisition mass spec. The authors compared proteomic profiles of high-purity EVs to washed crude EVs (both large and small) and determined that the higher-throughput crude EVs were very similar to the high-purity EVs, indicating that this isolation method may be sufficient and less time-consuming for biomarker discovery.

Major concerns:

1. In the methods "Serum sample preparation" section, the authors should provide more detail on how these steps were executed. Can an approximate range of blood or serum collected from patients be reported? How many participants provided blood and how many of those were OAP vs. healthy controls? I believe the number of OAP patients is mentioned in the results section but it would be helpful to include in this section of the methods.
2. Were serum samples pooled for the "Isolation of large and small extracellular vesicles from serum" portions of the work? The authors allude to this as they state that "A total of 20 ml cell-free serum was required...". Were all samples pooled and aliquoted at 20 mL or was each donor able to provide enough blood that 20 mL serum could be obtained from one person?
3. Can the authors clarify why an Optiprep density gradient was used to purify LEVs and a sucrose density-gradient was used to purify SEVs? Further, why were 480 uL fractions collected from the Optiprep gradient, but 500 uL fractions collected from the sucrose?
4. The authors show the presence of several EV markers but do not show the absence of any negative EV markers. MISEV 2018 (Thery, Witwer, et al., 2018) encourage the use of negative EV markers to show whether vesicle prep was contaminated or not. Can the authors please include one or two of these?
5. I wonder if the large amount of platelet, coagulation, and intracellular related proteins observed in the samples are due to ex vivo release/cellular lysis occurring during blood coagulation for serum collection. Additionally, the OAP samples had large amounts of insulin receptor and I am curious if this could be due to red blood cell lysis during blood coagulation, especially since red blood cells increase significantly during pregnancy. Could the authors please comment on these potential confounding issues.

Minor concerns:

1. The authors use samples from OAP patients but do not describe what this is. Could the authors please provide a very brief summary of this in the introduction so the reader is a bit more orientated to the disorder and why it might be important to study/how EVs could be affected.
2. Throughout the document the authors refer to their samples as 'plasma/serum', however serum was the only blood product utilized in this experiment. Please change the writing to reflect this, as plasma and serum are very different fluids.
3. In the last paragraph of the introduction, the authors state that "EVs derived from serum are largely composed of cellular proteins and membrane proteins." Can the authors provide a reference for this.
4. Can the authors please clarify if the "Isolation procedure for high-purity SEVs (HPSEVs)" utilized the supernatant from the HPLEVs? As it is written it almost seems that a different sample set of 20 mL cell-free serum was processed to obtain HPSEVs but I imagine that is not the case?

5. Why was the crude SEV pellet washed with 60 mL PBS prior to a second round of ultracentrifugation at 100,000 x g for the HPSEV isolation? This seems like a very large volume of PBS?
6. Can the authors please provide more detail for the methodology of "Sample preparation for Mass spectrometry". How many/what volume of EVs were lysed in 20 uL RIPA? What volumes or concentrations of Tris phosphine hydrochloride, iodoacetamide, and trypsin were used? Were these steps performed at room temperature?
7. Some of the arrows and labels in Figure 1 are crowded/overlapped. Please ensure proper spacing for all of these.
8. Can the authors clarify which fractions from the gradients were used for proteomic analysis of the high purity EVs?
9. The annotations in figure 3c are very difficult to read due to their size.
10. Was the immunodepleted serum purchased or was this depletion carried out in the lab? What proteins were depleted from the serum prior to analysis?

Reviewer #3 (Remarks to the Author):

This work introduced a deep proteome profiling workflow for circulating extracellular vesicles (EVs) based on diaPASEF technique. The authors designed two sample preparation approaches for either obtaining highly purified EVs for deep profiling or efficient EV purification for application to high-throughput cohort research. This method enabled large numbers of proteins identified in both small and large EVs in serum samples, which can serve as a powerful tool for clinical studies. Before the publication of this work, I suggest that the authors should first address a few major issues in their manuscript as follows.

1. In the 4th paragraph of the introduction section, you have introduced the development of proteome profiling methods in the past years. However, there still lacks a review of the latest and more related work, such as (1) and (2). Especially in (2), the authors also established a DIA-MS based workflow that is able to routinely and reproducibly quantify more than two thousand proteins in plasma EV samples. In addition, they have also optimized the sample preparation method to enable the analysis of hundreds of samples per day for high-throughput cohort study. What I am interested in is not only a review of related work, but also a more comprehensive comparison (better if you could provide evidence from experiments) to demonstrate the novelty of your method. Moreover, the statement "At present, the depth of protein identification..." is not precise enough in terms of what kinds of methods you want to include in your conclusion. Try not to use umbrella words like "in most large cohort studies", and try to cite concrete related papers.

(1) Muraoka S, Hirano M, Isoyama J, et al. Comprehensive proteomic profiling of plasma and serum phosphatidylserine-positive extracellular vesicles reveals tissue-specific proteins[J]. *Iscience*, 2022, 25(4).

(2) Kverneland A H, Østergaard O, Emdal K B, et al. Differential ultracentrifugation enables deep plasma proteomics through enrichment of extracellular vesicles[J]. *Proteomics*, 2023, 23(7-8): 2200039.

2. As you adopted a library-based DIA-MS analysis method, protein identification and quantification are highly related to the quality of your spectral library. I suggest that you provide more details about the library you built and have an evaluation of how many proteins can be identified and quantified and

how many cannot.

3. In Figure 1c, why there are two peaks for HPLEV? And it seems there's a significant overlap between HPLEV and HPSEV, so I'm wondering if there's a real need to characterize SEV and LEV separately. Or is there additional room to improve the purity of LEV and SEV?

4. If we compare Figure 1d and Figure 2d, the ranks and intensities of some of the biomarkers can change a lot, for example, Flotillin-1. Why there are such rank changes using different sample preparation approaches? Does this mean the two methods will lead to different quantification results and biological conclusions?

5. In Figure 1f and Figure 2e, the protein compositions within LEV exhibit notable similarity. However, the pattern of SEV proteins displays significant variability, with a particularly pronounced shift towards the dominant representation of secreted proteins in WCSEV. Why does this happen?

6. The resolution of Figure 3c is too low to see anything.

7. In Figure 4f, the highlighted pathways are not the top enriched ones for KEGG, which can be a biased result. Can you also explain more about the un-highlighted pathways?

8. In Figure 6, there are 3 levels for each aspect in the radar chart. I wonder how you "quantify" each item to be 1, 2 or 3?

There are a few minor issues that also need to be addressed as follows.

1. Why did you set the XIC IM extraction window to 0.8?

2. Please add citations to this statement in Paragraph 1 of Results section. "The diameters, measured by NanoSight technology (NTA), were about 140 nm for HPSEVs and 270 250-500 nm for HPLEVs, in accordance with previous reports."

3. I suggest using vector images instead of pixel images to improve the resolution.

4. I suggest also including data collection and analysis details in the Method section in the manuscript.

5. In the "Data Analysis" section of the reporting summary, there's a typo where you might have stated the same thing twice.

Response to Reviews

We are very grateful for the efforts that the reviewers spent on our manuscript. We sincerely appreciate the reviewer's recognition of our work. We have carefully gone through the comments and found them very helpful. Please find below our point-by-point responses. We have also revised the manuscript according to the reviewers' suggestions.

Reviewer #1:

Dear Tian and co-authors,

In this study, you have performed a Deep proteomic analysis of circulating large/small extracellular vesicles in patients with obstetric antiphospholipid syndrome. You performed a fascinating and comprehensive study, and the article contains new information. However, in some parts, it could be written a little more carefully. Some information is missing, some numbers do not agree... It is often not clear whether the authors are comparing EVs isolated by different methods (WCLEV vs. WCSEV, HPSEV vs. WCSEV, HPSEV vs. HPSEV), all isolated from healthy individuals, or whether they are comparing healthy EVs with OAPS EVs... I suggest that the authors clearly state throughout the paper what they are comparing and which cohorts were used for which analysis.

Response: We sincerely appreciate the reviewer's recognition of our work. As the reviewer stated, we first established procedures for isolation EVs or EV enriched fractions in Figure 1-3, with all samples being derived from healthy individuals.

We compared proteome of EVs isolated by two distinct methods: HPLEV (high-purity large EVs) and HPSEV (high-purity small EVs) in Figure 1 (line 258-267). For this, we employed density-gradient centrifugation to purify EVs from serum. We aimed to demonstrate that serum EVs contain a large repertoire of proteins, which is worthy of in-depth characterization in cohort studies using serum samples. Due to the complexity of obtaining HPLEV and HPSEV, we further develop methods to achieve a simple and deep characterization of EV protein compositions.

Subsequently, using the procedures depicted in Figure 2a (lines 320-322), we procured serum SEV and LEV enriched fractions, denoted as WCSEV and WCLEV respectively. A comparative proteomic analysis of these enriched fractions can be observed in Figures 2b-e and Figure 3.

In Figures 4 and 5, we incorporated this novel method to analyze a disease cohort, which included serum samples from both healthy individuals and OAPS patients (lines 394-397).

We are sorry for this misunderstanding and we deleted "CLEV and CSEV" both in figures and in manuscript. We also modified the figure legends to make it clearer.

Main comments:

1. Title. There is no information about the study population; this may broaden the scientific audience that will read the paper.

Response: We thank the reviewer for this point. In response, we have revised the title to

“Deep proteomic analysis of obstetric antiphospholipid syndrome by DIA-MS of large and small circulating extracellular vesicle enriched fraction”

2. Introduction. Add information about the size of the EVs at the end of the third paragraph (lines 83-84). I suggest that the first part of the results (lines 250-256) be moved to the introduction. This information should be available to readers at the very beginning.

Response: We thank the reviewer for this point. We moved the first part of the results (lines 250-256) to the introduction (line 84) and made a few modifications. Additionally, we included the size of the EVs at line 84.

3. Methods.

Response: We thank the reviewer for the valuable comments on the method section and we modified the manuscript accordingly. Specifically, the inclusion criteria for the cohort sample have been incorporated in the revised version. Additionally, we have restructured the order of the methods sections to align with the sequence presented in the text.

a. The order of the subtitles is illogical... it starts with antibody? Maybe: 1. study population, 2. sample preparation, 3. methods....., antibodies, 4. statistics...

Response: We change the order of subtitles in method section accordingly and added “study population” section with inclusion criteria of the cohort sample in the revised manuscript.

Study population:

The participants were recruited from the Medical Examination Center of Peking University Third Hospital in 2022. Female participants diagnosed with OAPS met the Sydney criteria as follows: (1) Three or more consecutive spontaneous abortions prior to the 10th week of gestation; (2) One or more unexplained deaths of a morphologically normal fetus at or beyond the 10th week of gestation with normal fetal morphology documented by ultrasound or by direct examination of the fetus; (3) One or more premature births of a morphologically normal neonate before the 34th week of gestation because of eclampsia or severe preeclampsia or recognized features of placental insufficiency. Participants were excluded if there were maternal anatomical or hormonal abnormalities, or if paternal and maternal chromosomal abnormalities caused abortion¹. Meanwhile, healthy age-matched women were selected as controls. A total of 24 healthy individuals and 20 OAPS were enrolled (Table 10). Demographics (age, gender, clinical characteristics) were obtained for all patients. In addition, 12 healthy individuals and 24 OAPS patients were enrolled for validation study (Table 9). The study was approved by the Ethics Committee of Peking University Third Hospital (ethical approval no. M2022836).

b. The description of the western blot analysis is missing

Response: We added the description of the western blot analysis in the method section.

Purified EVs were lysed by RIPA (Thermo, Cat. No. 89901) buffer for 2 hours, then centrifuged at 14,000 x g for 20 min. Supernatant with the same amount of protein were subjected to SDS loading buffer and boiled for 10-20 min at 95 °C. Samples were separated on SDS-PAGE gels of an appropriate percentage according to the MW of target proteins, followed by electrophoretic transfer onto Nitrocellulose membrane. Membranes were blocked with 5% non-fat milk in TBST buffer, and incubated with primary antibody

overnight at 4 °C. Membranes were incubated with HRP-conjugated secondary antibody for 1 hr at room temperature, and signals were detected with ECL Reagent (LABLEAD, E1050).

c. Serum sample preparation. Serum samples should be prepared promptly after blood collection to minimize the degradation of EVs and possible alterations. The authors did not specify how long they stored the serum at -80°C before storage

Response: All serum samples were prepared within 2 hours after blood collection. We added this information to the “Serum sample preparation” of method section.

d. Line 117: Serum was centrifuged at 3000 rpm... RPM units should be replaced by centrifugation force (\times g). As is stated in the rest of the manuscript.

e. I recommend using (\times g) units instead of just (g). For example, 1000 \times g, instead of 1000g

Response: We changed 3000 rpm...to 900 \times g in line 117. We also changed “g” to “ \times g” in the revised manuscript.

f. Please include '(Figure 1a)' in the paragraphs isolation procedures HPLEVs and HPSEVs. I suggest at the end of the first or second sentence, i.e.: "A total of 20 ml of cell-free serum was needed to purify a batch of HPLEVs (Figure 1a).

Response: We cited Figure 1a in Line 123 and Line 138 in the revised manuscript.

g. There is a missing paragraph about the participants included in the study, e.g.: 'Participants and clinical characteristics'. Please indicate that (or if) the workflow was first established using serum from healthy participants....(n=) and then add patient information. Please include information on inclusion criteria, how many patients were enrolled in the study, and which cohort was used for further analysis.

Response: We apologize for any confusion. The workflow was established using serum from healthy participants in Figure 2 (n=6). Serum of healthy individuals were only used in Figure 1-3 but not in the following cohort study. For OAPS disease cohort study, serum samples from OAPS patients as well as age-matched healthy individuals were analyzed (WCLEV: n=24 for healthy controls and n=20 for OAPS; WCSEV: n=18 for healthy controls and n=20 for OAPS) (Table 10). We also added “study population” in the method section to describe the “Participants and clinical characteristics”. We also added table 10 to include this information.

4. Results:

a. The first part of the results (lines 250-256) are actually not the results, as suggested this part should be in the introduction.

Response: We made the changes according to the reviewer’s suggestion.

b. Lines 258-267 repeat the methods. This part should be much shorter. It lacks details about the participants... Therefore, later in line 273 should be: "...and 5636 and 4145 proteins were identified...." It is not clear if this was in patient sera or healthy sera.

Response: We apologize for this ambiguity. Figure 1-3 depicted proteome of EVs derived from healthy individuals. These serum samples were only utilized for method development for Figure 1-3. The purpose of Figure 1-3 is to demonstrate the diverse protein repertoire

within serum EVs and subsequently establish a workflow that facilitates a streamlined yet comprehensive characterization of EV protein content, making it suitable for disease cohort studies.

Figure 4,5 depicted the analysis of serum samples from OAPS patients and matched healthy controls.

Line 258-267 did not involve OAPS participants but only serum samples of healthy individuals. We added “study population” section with inclusion criteria of the cohort sample. “and 5636 and 4145 proteins were identified....” was in healthy serum which was only used for method development.

We also simplified the paragraph of Line 258-267.

c. Line 396. Add specific information, how many, what age...? (HBD, n=57, mean age 31.2 years) (OAPS, n=46, mean age 31.3 years)

Response:

We added this information in the manuscript according to the reviewer’s suggestion. We also added table 10 to include this information.

For WCLEV analysis:

Healthy controls, n=24, age=33.25±4.93

OAPS, n=20, age=32.15±3.79

For WCSEV analysis:

Healthy controls, n=18, age=33.72±4.86

OAPS, n=20, age=32.15±3.79

d. Line 397: "44 cases" Which cases? In Table 6, there are 57 healthy individuals and 45 OAPS...?

Response:

We apologize for any confusion. “Line 397: Table 6 included information for Figure 5b. "44 cases" means 44 serum samples including healthy controls and OAPS were subjected for WCLEV and WCSEV preparation. 6 samples failed the preparation of WCSEV. We also added table 10 to include this information.

For WCLEV analysis:

Healthy controls, n=24, age=33.25±4.93

OAPS, n=20, age=32.15±3.79

For WCSEV analysis:

Healthy controls, n=18, age=33.72±4.86

OAPS, n=20, age=32.15±3.79

5. Discussion: line 567. did the authors investigate if there are some OAPS patients with a history of thrombosis in their study? Maybe they could check if these patients have higher VWF values in their EVs.

Response: In terms of the clinical diagnostic criteria, all patients were diagnosed with APS with obstetric adverse events, and none had a history of thrombosis, according to a review of

their medical records. In addition, the D-Dimer level of OAPS patients is within the normal range (Table 6). The increased level of VWF in WCLEV in OAPS patients may indicate tendency of hypercoagulability in OAPS patients, which raise more attention to the early hint of thrombosis in OAPS patients.

Reviewer #2:

In this manuscript the authors describe a novel approach to establishing deep proteomic profiling of serum EVs via 4D-data independent acquisition mass spec. The authors compared proteomic profiles of high-purity EVs to washed crude EVs (both large and small) and determined that the higher-throughput crude EVs were very similar to the high-purity EVs, indicating that this isolation method may be sufficient and less time-consuming for biomarker discovery.

Major concerns:

1. In the methods “Serum sample preparation” section, the authors should provide more detail on how these steps were executed. Can an approximate range of blood or serum collected from patients be reported? How many participants provided blood and how many of those were OAP vs. healthy controls? I believe the number of OAP patients is mentioned in the results section but it would be helpful to include in this section of the methods.

Response: We sincerely appreciate the reviewer’s recognition of our work and thank the reviewer for these valuable comments.

We apologize for this ambiguity. To clarify, For Figure 1, 80 ml serum from 80 healthy controls were collected. For workflow development in Figure 2, 1 ml serum sample from 6 healthy controls were collected. For cohort study in Figure 4, a total of 24 healthy individuals and 20 OAPS were enrolled (Table 10). In Figure 5, serum samples from 12 healthy individuals and 24 OAPS were used (Table 9). The study was approved by the Ethics Committee of Peking University Third Hospital (ethical approval no. M2022836).

We added how these steps were executed in “Serum sample preparation” in method section. We also added Table 10 and cited in this section.

Serum sample preparation:

3 ml blood samples of each participant for research purposes were collected in pro-coagulation vacuum tubes using standard venepuncture protocols. These samples were centrifuged at 900 x g for 10 minutes to extract serum, which was then preserved at -80°C until further use. All serum samples were prepared within 2 hours after blood collection.

2. Were serum samples pooled for the “Isolation of large and small extracellular vesicles from serum” portions of the work? The authors allude to this as they state that “A total of 20 ml cell-free serum was required...”. Were all samples pooled and aliquoted at 20 mL or was each donor able to provide enough blood that 20 mL serum could be obtained from one person?

Response:

We apologize for this ambiguity. In Figure 1, a single purification of large and small extracellular vesicles by gradient centrifugation requires 20ml of serum. We conducted 4

independent purifications in Figure 1. Serum from 20 healthy controls (1ml of each person) were mixed for one purification. A total of 80ml serum was used.

3. Can the authors clarify why an Optiprep density gradient was used to purify LEVs and a sucrose density-gradient was used to purify SEVs? Further, why were 480 uL fractions collected from the Optiprep gradient, but 500 uL fractions collected from the sucrose?

Response:

Iodixanol serves as the primary constituent of the Optiprep density gradient. Both sucrose and iodixanol are frequently employed as gradient separation media for EV purification. The literature has detailed the use of either iodixanol or sucrose for SEVs. A prior study from our group delineated a technique for the purification of large extracellular vesicles from serum using the Optiprep density gradient, a methodology that we have adapted in this manuscript².

Further, why were 480 uL fractions collected from the Optiprep gradient, but 500 uL fractions collected from the sucrose?

We are sorry for the discrepancy. 480 uL fractions were collected for both gradients. The total volume of the density gradient was 5 ml and we collected 10 fractions, theoretically 500 uL for each fraction. Taking the volume loss during the experiments into account, we collected 480 uL per fraction to ensure equal volume of each fraction. We modified the volume in the revised manuscript.

4. The authors show the presence of several EV markers but do not show the absence of any negative EV markers. MISEV 2018 (Thery, Witwer, et al., 2018) encourage the use of negative EV markers to show whether vesicle prep was contaminated or not. Can the authors please include one or two of these?

Response:

In our revised figures, we have incorporated several markers conventionally recognized as negative exosome indicators, namely ER marker (Calnexin), and the Mitochondrial marker (Tom20) (Figure R1). These additions confirm that the exosomes purified in this study remained uncontaminated by intracellular organelles. Intriguingly, certain reports, supported by our findings, indicate that HPLEV does encompass intracellular components^{2,3} and performs specific functions. This suggests a new direction for LEV research. We have also made reference to the MISEV 2018 guidelines⁴.

Figure R1. Immunoblots showing the presence of indicated proteins in HPLEV or HPSEV, respectively. (Figure 1b in revised manuscript)

5. I wonder if the large amount of platelet, coagulation, and intracellular related proteins observed in the samples are due to ex vivo release/cellular lysis occurring during blood coagulation for serum collection.

Response:

We thank the reviewer for this good point and we had taken it into account. Several studies, supported by a growing body of evidence, have confirmed that circulating extracellular vesicles (EVs) encompass a vast array of intracellular components. Therefore, the detection of proteins related to intracellular pathways in circulating EVs is anticipated. For sample preparation, we meticulously followed a rigorous Standard Operating Procedure (SOP) designed to minimize the risk of cellular lysis. Serum samples were prepared within a 2-hour window. We systematically excluded any serum samples exhibiting visible hemolysis. It's essential to note that all samples were consistently processed according to this SOP, suggesting that any cellular lysis, if it occurred, would be minimal and consistent across samples. While analyzing serum samples, it is pertinent to acknowledge the potential for some ex vivo release during coagulation. This is an inherent limitation not only specific to EV proteomics but blood proteomics. Even though such ex vivo releases would be uniformly present across all samples, potentially masking variations in protein concentrations, our analysis still revealed discernible differences in protein abundance between healthy controls and OAPS patients. This underscores that a considerable number of proteins are available in our method to be associated with this disease. We concur with the reviewer's perspective that pre-analytical variance should be considered in data interpretation.

Additionally, the OAP samples had large amounts of insulin receptor and I am curious if this could be due to red blood cell lysis during blood coagulation, especially since red blood cells increase significantly during pregnancy. Could the authors please comment on these potential confounding issues.

It's worth noting that serum samples from OAPS patients in our study were collected one month after the abortion, and there are no indications of elevated red blood cell counts of the patients. Given the cellular origin of the vesicular component, we postulate that the variance discerned in the insulin receptor is indicative of specific functional abnormalities manifesting within the WCSEV fraction.

Minor concerns:

1. The authors use samples from OAP patients but do not describe what this is. Could the authors please provide a very brief summary of this in the introduction so the reader is a bit more orientated to the disorder and why it might be important to study/how EVs could be affected.

Response:

We thank the reviewer for this good point. We added a brief introduction of OAPS in the result part. We hope it will better help the reader understand how EVs proteomics could provide new insight in this disease.

Line 394 of result part:

Antiphospholipid syndrome (APS) is an autoantibody-induced thrombophilia, frequently manifesting as recurrent thrombosis and pregnancy complications. The predominant pathogenesis involves endothelial cells, monocytes, platelets, and complement in the onset of thrombosis and fetal death in antiphospholipid syndrome⁵. Obstetric APS (OAPS) is featured by recurrent early miscarriages, fetal death at or beyond 10 weeks of gestation, and early delivery due to severe preeclampsia or placental insufficiency¹.

2. Throughout the document the authors refer to their samples as ‘plasma/serum’, however serum was the only blood product utilized in this experiment. Please change the writing to reflect this, as plasma and serum are very different fluids.

Response:

We thank the reviewer for this good point. We have changed “plasma/serum” to “serum” in line 256-257 and line 525-528.

3. In the last paragraph of the introduction, the authors state that “EVs derived from serum are largely composed of cellular proteins and membrane proteins.” Can the authors provide a reference for this.

Response:

We apologize for this ambiguity. This sentence is a description of the finding in this manuscript. And it is consistent with the finding in other literatures. We have added the literature.

(1) Whitham, M. & Febbraio, M. A. Redefining Tissue Crosstalk via Shotgun Proteomic Analyses of Plasma Extracellular Vesicles. *Proteomics* 19, e1800154, doi:10.1002/pmic.201800154 (2019).

(2) Whitham, M. et al. Extracellular Vesicles Provide a Means for Tissue Crosstalk during Exercise. *Cell Metab* 27, 237-251 e234, doi:10.1016/j.cmet.2017.12.001 (2018).

4. Can the authors please clarify if the “Isolation procedure for high-purity SEVs (HPSEVs)” utilized the supernatant from the HPLEVs? As it is written it almost seems that a different sample set of 20 mL cell-free serum was processed to obtain HPSEVs but I imagine that is not the case?

Response:

We apologize for this ambiguity. Indeed, the “Isolation procedure for high-purity SEVs (HPSEVs)” utilized the supernatant from the HPLEVs. In line 141, we have changed the description to “The serum supernatant after 20,000 x g centrifugation was further centrifuged at 100, 000 x g for 70min at 4 °C”.

5. Why was the crude SEV pellet washed with 60 mL PBS prior to a second round of ultracentrifugation at 100,000 x g for the HPSEV isolation? This seems like a very large volume of PBS?

Response:

We apologize for the lack of clarity. To elaborate, we diluted a 20 ml serum sample into 120 ml, which was then divided between two centrifuge tubes for ultracentrifugation. Post-centrifugation, the pellets were re-suspended in 60 ml PBS and combined into a single tube for a subsequent ultracentrifugation round. The increased PBS volume enhances the washing step. This description has been updated in line 140 of the methods section.

6. Can the authors please provide more detail for the methodology of “Sample preparation for Mass spectrometry”. How many/what volume of EVs were lysed in 20 uL RIPA? What volumes or concentrations of Tris phosphine hydrochloride, iodoacetamide, and trypsin were used? Were these steps performed at room temperature?

Response:

We are sorry for not making this clear. The protein amount of WCLEV and WCSEV from 1 ml serum was approximately 5-20 μg . The concentration of Tris phosphine hydrochloride is 10 mM, and iodoacetamide is 25 mM. 1 μg trypsin was added to the reaction. All procedures conducted at room temperature except trypsin digestion. We have added this information to “Sample preparation for Mass spectrometry”.

7. Some of the arrows and labels in Figure 1 are crowded/overlapped. Please ensure proper spacing for all of these.

Response:

We are sorry for this and we modified Figure 1 in the revised version.

8. Can the authors clarify which fractions from the gradients were used for proteomic analysis of the high purity EVs?

Response:

We are sorry for not making this clear. For HPLEV, the 4 and 5 fractions from the top (480 μl /fraction) were collected for proteomic analysis. For HPSEV, the 8 and 9 fractions from the top (480 μl /fraction) were collected for proteomic analysis. For the reviewer’s convenience, we presented the pictures we took after gradient centrifugation (Figure R2).

Figure R2. Pictures after gradient centrifugation: Left panel, HPLEV, right panel, HPSEV. Red arrow head pointed to the fractions which were collected for proteomic analysis.

9. The annotations in figure 3c are very difficult to read due to their size.

Response:

We are sorry for this. We changed to high resolution figure in the revised version.

10. Was the immunodepleted serum purchased or was this depletion carried out in the lab? What proteins were depleted from the serum prior to analysis?

Response:

We apologize for the ambiguity. The protein depletion was executed in our laboratory utilizing the High-Select™ Top14 Abundant Protein Depletion Resin. The 14 targeted proteins included: Albumin, IgA, IgD, IgE, IgG, IgG (light chains), IgM, Alpha-1-acid glycoprotein, Alpha-1-antitrypsin, Alpha-2-macroglobulin, Apolipoprotein A1, Fibrinogen, Haptoglobin, and Transferrin. Since the data referenced is from our previous publication⁶ and was solely cited for analytical purposes, we initially omitted the “Depletion of Highly Abundant Proteins from Serum Samples” from the methods section. However, for clarity and the convenience of the

reviewer, we are providing a detailed description of the procedure below:

Depletion of Highly Abundant Proteins from Serum Samples

High-Select Top14 Abundant Protein Depletion of serum samples was conducted according to the manufacturer instructions of “Multi Affinity Removal Column, Human-14” (Agilent Technologies). Briefly, 20 μ L of serum and 60 μ L of buffer A were mixed evenly, transferred into filter 0.22 μ m cellulose acetate certified tube (COSTOR), and subsequently centrifuged at $10,000 \times g$ for 1 min. The filtrate was removed into 29 \times 5 mm preassembled plastic springs for High-Select Top14 Abundant Protein Depletion using a liquid chromatography system (Agilent Technologies 1290). The depleted samples were collected and the protein concentration was measured using a BCA Protein Assay Kit (Thermo Scientific) following the manufacturer’s instructions.

Reviewer #3:

This work introduced a deep proteome profiling workflow for circulating extracellular vesicles (EVs) based on diaPASEF technique. The authors designed two sample preparation approaches for either obtaining highly purified EVs for deep profiling or efficient EV purification for application to high-throughput cohort research. This method enabled large numbers of proteins identified in both small and large EVs in serum samples, which can serve as a powerful tool for clinical studies. Before the publication of this work, I suggest that the authors should first address a few major issues in their manuscript as follows.

Response: We sincerely appreciate the reviewer’s recognition of our work.

1. In the 4th paragraph of the introduction section, you have introduced the development of proteome profiling methods in the past years. However, there still lacks a review of the latest and more related work, such as (1) and (2). Especially in (2), the authors also established a DIA-MS based workflow that is able to routinely and reproducibly quantify more than two thousand proteins in plasma EV samples. In addition, they have also optimized the sample preparation method to enable the analysis of hundreds of samples per day for high-throughput cohort study. What I am interested in is not only a review of related work, but also a more comprehensive comparison (better if you could provide evidence from experiments) to demonstrate the novelty of your method. Moreover, the statement “At present, the depth of protein identification...” is not precise enough in terms of what kinds of methods you want to include in your conclusion. Try not to use umbrella words like “in most large cohort studies”, and try to cite concrete related papers.

(1) Muraoka S, Hirano M, Isoyama J, et al. Comprehensive proteomic profiling of plasma and serum phosphatidylserine-positive extracellular vesicles reveals tissue-specific proteins[J]. *IScience*, 2022, 25(4).

(2) Kverneland A H, Østergaard O, Emdal K B, et al. Differential ultracentrifugation enables deep plasma proteomics through enrichment of extracellular vesicles[J]. *Proteomics*, 2023, 23(7-8): 2200039.

Response:

We thank the reviewer for this valuable point. We have compiled a comparison in Table R1 to elucidate the distinctiveness and advantages of our method. We also cited these two papers in the revised manuscript. We also deleted the sentence from Line 90-92.

	Pre-analytical Methods	Depth of identification	Volume (µl)	Distinguish LEV/SEV	MS method	Large sample cohort	Validation
Muraoka S et al	affinity capture	3852 (serum EV)/ 3791(plasma EV)	150	Mixture of LEV and SEV	3D-DIA	No	No
Kverneland A H	ultracentrifugation (One-step wash)	2331(LEV)	1000	LEV only	3D-DIA	No	No
Our manuscript	ultracentrifugation (Two-step wash)	4270 (LEV) /3328 (SEV)	1000	Both LEV and SEV	4D-DIA	Yes	Yes

It is worth noticing that:

(1) A shared observation between Muraoka S et al., Kverneland A H, and our study is the following: enriching for EV fractions markedly enhances the depth of serum proteomic identification. This accentuates that serum EV proteomics is a potent and underexplored reservoir of protein data warranting further exploration.

(2) Notably, neither of the aforementioned studies differentiates between small EVs enriched fractions and large EVs enriched fractions. Small EVs, often classified as exosomes, and large EVs, often microvesicles – have inherent attributes. While exosomes stem from endocytic pathways, microvesicles arise from the outward budding of vesicles off the plasma membrane. Consequently, they possess discrete surface proteins and encapsulated cargo, thereby orchestrating distinct biological roles. The concurrent procurement of quantitative proteomic data from both these enriched EV factions augment our comprehension of serum proteomics.

(3) Moreover, neither of the aforementioned studies have replicated their proteomic analysis methodologies on EV fractions in serum across an expansive set of both healthy and diseased individuals. In contrast, our study has confirmed the applicability of the approach delineated in our manuscript across 44 serum samples from a specific disease cohort. Furthermore, our study underscores its validation process.

2. As you adopted a library-based DIA-MS analysis method, protein identification and quantification are highly related to the quality of your spectral library. I suggest that you provide more details about the library you built and have an evaluation of how many proteins can be identified and quantified and how many cannot.

Response:

We are sorry for not making this clear.

For the DIA-proteomic data presented in Figure 1, a spectral library from the analyzed samples were used for DIA data identification. Meanwhile, for the proteomic data in Figures 2-5, a hybrid of sample-specific spectral library and direct search-based library were employed for DIA data identification. For the spectral library generation, a 1% FDR was maintained at the PSM, peptide, and protein levels. The raw data of the library we built has been have been deposited in iProX (integrated proteome resources) of ProteomeXchange with the accession code PXD043290. We also added this in the method section.

3. In Figure 1c, why there are two peaks for HPLEV? And it seems there's a significant overlap between HPLEV and HPSEV, so I'm wondering if there's a real need to characterize SEV and LEV separately. Or is there additional room to improve the purity of LEV and SEV?

Response:

We thank the reviewer for pointing this out. Consistent with the literature, microparticles exhibit a diameter range of 100 nm-1 μ m, reflecting significant size heterogeneity. Our findings mirror these observations. For HPLEV isolation from serum, the fraction from 20,000 x g centrifugation was followed by density-gradient centrifugation. Measured by NanoSight technology (NTA), HPLEV diameters ranged between 250-500 nm, with prominent peaks observed at 250 nm and 450 nm. This heterogeneity was further confirmed through electron microscopy. Since, vesicle size, contents and gravity are not necessarily correlated perfectly, it is possible that the 20,000 x g fraction encompasses vesicles of diverse diameters.

It is necessary to characterize SEV and LEV separately. They harbor distinct proteins and exert different biological functions. In cohort applications, it is necessary to characterize SEV and LEV enriched fractions (WCSEV/WCLEV) separately. WCSEVs and WCLEVs are isolated based on their gravity and shape using different centrifugal force (20,000 x g for LEV and 100,000 x g for SEV), thus they are different vesicles enriched fractions. Moreover, as presented in the manuscript, only 60 DEPs overlap between WCLEVs and WCSEVs, which suggests that WCLEVs and WCSEVs carry different disease-related information.

While vesicle purity can always be enhanced—potentially through a combination of strategies such as ultracentrifugation, gradient centrifugation, and affinity purification⁷. These techniques might lead to lower throughput and demand a substantial initial sample volume, a significant increase in cost, and so on. A balanced consideration of throughput, required sample quantity, cost, and purity is crucial for cohort study. As evidenced, our established workflow demonstrates feasibility for cohort studies.

4. If we compare Figure 1d and Figure 2d, the ranks and intensities of some of the biomarkers can change a lot, for example, Flotillin-1. Why there are such rank changes using different sample preparation approaches? Does this mean the two methods will lead to different quantification results and biological conclusions?

Response:

The reviewer raised a very good point. Comparing the biomarker rankings between HPLEV/HPSEV (Figure 1d) and WCLEV/WCSEV (Figure 2d), flotillin-1's rank shifted dramatically from 275 to 899. As elaborated in our manuscript, HPLEV/HPSEV and WCLEV/WCSEV are not entirely synonymous, with the latter being an LEV/SEV enriched fraction. The procedures to procure WCLEV or WCSEV might co-purify certain proteins, impacting protein rankings. However, evaluating commonly recognized proteins like CD81, CD63, TSG101, Alix, Syntenin-1, Integrin β 1, Annexin A2, and Annexin A5 reveals a roughly consistent ranking. Thus, the WCLEVs/WCSEVs analyzed in our workflow are EV proteins enriched, which are largely representative of EV proteins. These two methods may lead to different quantification results. Both of them could represent disease features. The biological conclusions overlapped to some extent.

5. In Figure 1f and Figure 2e, the protein compositions within LEV exhibit notable similarity. However, the pattern of SEV proteins displays significant variability, with a particularly pronounced shift towards the dominant representation of secreted proteins in WCSEV. Why does this happen?

Response:

We thank the reviewer for pointing this out. We have indeed recognized this observation and

have provided an extensive discussion on it within the Discussion section of our manuscript. It is noteworthy that both LEV and SEV incorporate intracellular components that are of medium to low abundance. We postulate that the presence of copurified proteins could modulate the mass spectrometry analysis of the samples, consequently affecting the observed compositions. We also cited the discussion for the reviewer's convenience.

In this study, we noticed larger discrepancy in protein identification between HPSEVs (4145) and WCSEVs (1542) than between HPLEVs (5636) and WCLEVs (4000). Our workflow to obtain WCLEV or WCSEV fractions from serum inevitably involved in copurified proteins. It is highly possible that the 100,000g centrifugation step in the preparation of WCSEV resulted in different highly abundant copurified proteins compared to the 20,000g centrifugation step in the purification of WCLEV, which influenced their protein identification by mass spectrometry. For WCSEVs, the top 10 abundant proteins are mostly secreted proteins. They are immunoglobins, complement proteins, apolipoproteins and albumin, and they account for more than 40% of the total protein amount. In contrast, the top 10 abundant proteins in WCLEVs are mostly intracellular proteins, such as actin, actin-regulated proteins and integrins with relatively less albumin and immunoglobins. They account for about 30% of the total protein amount.

6. The resolution of Figure 3c is too low to see anything.

Response:

We are sorry for this. We changed to high resolution figure in the revised version.

7. In Figure 4f, the highlighted pathways are not the top enriched ones for KEGG, which can be a biased result. Can you also explain more about the un-highlighted pathways?

Response:

We are sorry for not making this clear. In Figure 4f, the highlighted KEGG or GO pathways have been previously identified by other studies using samples such as serum/plasma, fibrin clots, platelets, and monocytes. For instance, the KEGG pathways "platelet activation, regulation of actin cytoskeleton" are delineated by a purple box, indicating that these pathways have been identified in studies using platelet samples. The purpose of Figure 4f is to show that proteomics of EV enriched fractions can offer a multi-encompassing perspective on disease pathogenicity, comparable to insights derived from multiple other sample types. Interestingly, the GO pathways highlighted in orange box demonstrated the consistency of EV proteomics data with findings from monocyte studies. The un-highlighted pathways were more discussed in Figure 5.

To make the manuscript clear, we added a sentence in line 414:

The color-coded boxes in Figure 4f correspond to the OAPS-associated KEGG and GO pathways identified in platelets, fibrin clot, plasma and serum, and monocyte samples, respectively, as documented in previous literature. Thus, proteomics of EV enriched fractions can offer a multi-encompassing perspective on disease pathogenicity, comparable to insights derived from multiple other sample types.

8. In Figure 6, there are 3 levels for each aspect in the radar chart. I wonder how you "quantify" each item to be 1, 2 or 3?

Response:

We acknowledge that we lack quantitative data for each item. The intention behind the figure was solely illustrative, aiming to depict the advantages associated with different methodologies. We could remove the figure in the revised manuscript.

- 1 Alijotas-Reig, J. *et al.* Pathogenesis, Diagnosis and Management of Obstetric Antiphospholipid Syndrome: A Comprehensive Review. *J Clin Med* **11**, doi:10.3390/jcm11030675 (2022).
- 2 Zhao, X. *et al.* Identification of markers for migrasome detection. *Cell Discov* **5**, 27, doi:10.1038/s41421-019-0093-y (2019).
- 3 Jiao, H. *et al.* Mitocytosis, a migrasome-mediated mitochondrial quality-control process. *Cell* **184**, 2896-2910 e2813, doi:10.1016/j.cell.2021.04.027 (2021).
- 4 They, C. *et al.* Minimal information for studies of extracellular vesicles 2018 (MISEV2018): a position statement of the International Society for Extracellular Vesicles and update of the MISEV2014 guidelines. *J Extracell Vesicles* **7**, 1535750, doi:10.1080/20013078.2018.1535750 (2018).
- 5 Ruiz-Irastorza, G., Crowther, M., Branch, W. & Khamashta, M. A. Antiphospholipid syndrome. *Lancet* **376**, 1498-1509, doi:10.1016/S0140-6736(10)60709-X (2010).
- 6 Chen, Y. *et al.* Proteomic Analysis Identifies Prolonged Disturbances in Pathways Related to Cholesterol Metabolism and Myocardium Function in the COVID-19 Recovery Stage. *J Proteome Res* **20**, 3463-3474, doi:10.1021/acs.jproteome.1c00054 (2021).
- 7 Shurtleff, M. J., Temoche-Diaz, M. M., Karfilis, K. V., Ri, S. & Schekman, R. Y-box protein 1 is required to sort microRNAs into exosomes in cells and in a cell-free reaction. *Elife* **5**, doi:10.7554/eLife.19276 (2016).

Reviewers' comments:

Reviewer #1 (Remarks to the Author):

Reviewer #2 (Remarks to the Author):

I would like to thank the authors for their careful consideration of my comments during the first round of review. I appreciate the thorough answers to all of my concerns. However there are a few additional items I would like the reviewers to address.

The authors use a combination of iodixanol and sucrose gradients for their EV separations. They use Optiprep/iodixanol to purify LEVs and sucrose density gradient to purify SEVs. I understand that both are commonly used in the field, however it is not clear why in this particular manuscript the authors chose to utilize both types of gradients. They have a protocol for purifying LEVs using Optiprep, why not use Optiprep for purification of SEVs as this is done in the literature?

I appreciate the photos included as Figure R2 to depict the fractions used for proteomic analysis. Can the authors please include clarification in the manuscript that fractions 4 and 5 of the HPLEV were used for proteomic analysis while fractions 8 and 9 were used for HSPEV?

Reviewer #3 (Remarks to the Author):

The authors have fixed most of the issues but there are still several minor concerns remaining to be addressed before publication as follows.

1. Note that the quality of the spectral library is a crucial factor for library-based DIA analysis workflow. Therefore, the details of the spectral libraries used are still not clear enough. Please consider the following issues. How many precursors, peptides and proteins can you obtain respectively from the DDA samples at 1% FDR? How many precursors, peptides and proteins can you identify and quantify respectively in DIA data? How is the protein coverage of your method? Why did you use different libraries in different experiments? How did you combine two libraries as you mentioned that you used a "hybrid library"?

2. I think the "were" in Line 249 and Line 250 should be "was".

3. The authors clarify in Line 289 that the result of NTA is in accordance with previous reports. Please cite specific publications. And is the "two-peak pattern" of HPLEV shown in Figure 1c also in accordance with previous work?

4. The font size of the annotations in Figure 3c still needs to be scaled up.

Response to Reviews

We are very grateful for the time and effort the reviewers have dedicated to our manuscript. We sincerely appreciate the acknowledgment of our initial revision. We have thoroughly addressed the points raised in the second review. Please find the point-to-point response according to the reviewers' suggestions.

Reviewer #1 (Remarks to the Author):

Response: We thank the reviewer for his recognition of our revision.

Reviewer #2 (Remarks to the Author):

I would like to thank the authors for their careful consideration of my comments during the first round of review. I appreciate the thorough answers to all of my concerns. However there are a few additional items I would like the reviewers to address.

Response: We are grateful for the reviewer's positive feedback and recognition of our efforts.

The authors use a combination of iodixanol and sucrose gradients for their EV separations. They use Optiprep/iodixanol to purify LEVs and sucrose density gradient to purify SEVs. I understand that both are commonly used in the field, however it is not clear why in this particular manuscript the authors chose to utilize both types of gradients. They have a protocol for purifying LEVs using Optiprep, why not use Optiprep for purification of SEVs as this is done in the literature?

Response: We thank the reviewer for this point. Actually, we did not have preference on choosing gradient reagents. The method used in this manuscript is adapted and modified from our previous study in which we chose Optiprep/iodixanol for LEVs and sucrose density gradient for SEVs¹. We believe both reagents are suitable for LEV and SEV purification.

I appreciate the photos included as Figure R2 to depict the fractions used for proteomic analysis. Can the authors please include clarification in the manuscript that fractions 4 and 5 of the HPLEV were used for proteomic analysis while fractions 8 and 9 were used for HSPEV?

Response: 4. We are pleased that the reviewer appreciates the presentation of Figure R2. We have added this information in Figure 1 as well as in materials and methods session.

Reviewer #3 (Remarks to the Author):

The authors have fixed most of the issues but there are still several minor concerns remaining to be addressed before publication as follows.

1. Note that the quality of the spectral library is a crucial factor for library-based DIA analysis workflow. Therefore, the details of the spectral libraries used are still not clear enough. Please consider the following issues. How many precursors, peptides and proteins can you obtain respectively from the DDA samples at 1% FDR? How many precursors, peptides and proteins can you identify and quantify respectively in DIA data? How is the protein coverage of your method?

Response: We thank the reviewer for this point. For the detailed number of libraries, please refer to the following table. We used Spectronaut for generating the spectral library.

DDA Library	Precursor	Peptides	Protein
WCLEV	115727	79275	6609
WCSEV	52526	38122	3997

For DIA file analyzing, detailed numbers for WCLEV and WCSEV from the cohort samples are provided in the following table:

Cohort DIA file hybrid search	Precursor	Peptides	Protein
WCLEV	68311	51500	4270
WCSEV	51217	36365	3328

Since the hybrid library search (a combination of direct search and spectral library search) was adapted in this manuscript, the identified proteins are not solely from searching the spectral library. Thus, it is not valid to show the coverage by comparing proteins identified from Hybrid library search spectral library. To address the reviewer's query, we compared the proteins identified from WCLEV and WCSEV in this manuscript to total proteins in the spectral library (Figure R1).

Figure R1. Venn diagram showing the overlapping proteins in the indicated samples.

Why did you use different libraries in different experiments?

WCLEV, WCSEV are different fractions extracted from serum with different protein compositions. To obtain accurate quantification of proteins in different type of samples, building spectral library for each type of sample individually is necessary as was performed in the previous study²⁻⁴.

How did you combine two libraries as you mentioned that you used a “hybrid library”?

Hybrid search is a data analysis feature by combining direct search and spectral library-based search in Spectronaut software's pulsar search engine. Please see detailed explanation from the link below:

http://files.biognosys.ch/058_Spectronaut/ReleaseMaterial/00_Manual/Spectronaut_17_UserManual.pdf

This method is widely used by others studies⁵.

2. I think the “were” in Line 249 and Line 250 should be “was”.

Response: We thank the reviewer for this point and we made the change accordingly in the revised manuscript.

3. The authors clarify in Line 289 that the result of NTA is in accordance with previous reports. Please cite specific publications. And is the “two-peak pattern” of HPLEV shown in Figure 1c also in accordance with previous work?

Response: We thank the reviewer for this point. We added the citation in the revised manuscript. Specifically, it has been reported that the diameter of LEV ranges from 100nm to 1 μm ⁶. In a research article “Isolation of exosomes from whole blood by integrating acoustics and microfluidics”, the author designed an acoustofluidic platform for extracellular vesicle subgroup separation⁷. Figure 4A as cited here displayed the size distribution of microvesicle with 3 peaks as 172nm, 263nm and 510nm. The data were obtained from NTA assays. Thus, the size of microvesicle is not evenly distributed, consistent with our findings.

Figure R2. Size distribution of microvesicle samples by NTA⁷.

4. The font size of the annotations in Figure 3c still needs to be scaled up.

Response: We thank the reviewer for this point. We scaled up the font size of the annotations in Figure 3c in revised figure.

Reference:

- 1 Zhao, X. *et al.* Identification of markers for migrasome detection. *Cell Discov* **5**, 27, doi:10.1038/s41421-019-0093-y (2019).
- 2 Chen, Y. *et al.* Proteomic Analysis Identifies Prolonged Disturbances in Pathways Related to Cholesterol Metabolism and Myocardium Function in the COVID-19 Recovery Stage. *J Proteome Res* **20**, 3463-3474, doi:10.1021/acs.jproteome.1c00054 (2021).
- 3 Chen, Y. *et al.* Immune response pattern across the asymptomatic, symptomatic and convalescent periods of COVID-19. *Biochim Biophys Acta Proteins Proteom* **1870**, 140736, doi:10.1016/j.bbapap.2021.140736 (2022).
- 4 Tian, W. *et al.* Immune suppression in the early stage of COVID-19 disease. *Nat Commun* **11**, 5859, doi:10.1038/s41467-020-19706-9 (2020).
- 5 Karayel, O. *et al.* Proteome profiling of cerebrospinal fluid reveals biomarker candidates for Parkinson's disease. *Cell Rep Med* **3**, 100661, doi:10.1016/j.xcrm.2022.100661 (2022).
- 6 They, C., Ostrowski, M. & Segura, E. Membrane vesicles as conveyors of immune responses. *Nat Rev Immunol* **9**, 581-593, doi:10.1038/nri2567 (2009).
- 7 Wu, M. *et al.* Isolation of exosomes from whole blood by integrating acoustics and microfluidics. *Proc Natl Acad Sci U S A* **114**, 10584-10589, doi:10.1073/pnas.1709210114 (2017).